# Bioarchitectonic Nanophotonics by Replication and Systolic Miniaturization of Natural Forms

**DOI:** 10.3390/biomimetics9080487

**Published:** 2024-08-13

**Authors:** Konstantina Papachristopoulou, Nikolaos A. Vainos

**Affiliations:** Photonics Nanotechnology Research Laboratory-PNRL, Department of Materials Science, University of Patras, 26504 Patras, Greece; vainos@upatras.gr

**Keywords:** bioarchitectonic, biomimetic, bioinspiration, photonics, optics, nanotechnology, aerogel, systolic, compound eye, ommatidia, nanoneedles, microtrichia

## Abstract

The mimesis of biological mechanisms by artificial devices constitutes the modern, rapidly expanding, multidisciplinary biomimetics sector. In the broader bioinspiration perspective, however, bioarchitectures may perform independent functions without necessarily mimicking their biological generators. In this paper, we explore such *Bioarchitectonic* notions and demonstrate three-dimensional photonics by the exact replication of insect organs using ultra-porous silica aerogels. The subsequent conformal systolic transformation yields their miniaturized affine ‘clones’ having higher mass density and refractive index. Focusing on the paradigms of *ommatidia*, the compound eye of the hornet *Vespa crabro flavofasciata* and the *microtrichia* of the scarab *Protaetia cuprea phoebe*, we fabricate their aerogel replicas and derivative clones and investigate their photonic functionalities. Ultralight aerogel microlens arrays are proven to be functional photonic devices having a focal length f ~ 1000 μm and f-number f/30 in the visible spectrum. Stepwise systolic transformation yields denser and affine functional elements, ultimately fused silica clones, exhibiting strong focusing properties due to their very short focal length of f ~ 35 μm and f/3.5. The fabricated transparent aerogel and xerogel replicas of *microtrichia* demonstrate a remarkable optical waveguiding performance, delivering light to their sub-100 nm nanotips. Dense fused silica conical clones deliver light through sub-50 nm nanotips, enabling nanoscale light–matter interactions. Super-resolution bioarchitectonics offers new and alternative tools and promises novel developments and applications in nanophotonics and other nanotechnology sectors.

## 1. Introduction

Nanotechnology in three-dimensional (3D) space opens many avenues for future spatiotemporal light–matter interactions in the molecular and atomistic domain. Even though advanced planar lithographic processing technologies enable fabrication in the sub-10 nm scale, building 3D functional structures in space remains a challenge.

Diffraction of light and particle waves imposes fundamental physical limitations on the resolution of lithographic processing and alternative new tools are needed to reach miniaturization limits and explore the inherent and far-reaching potential of 3D nanodevices.

Current technology employs two-dimensional (2D) planar processing methods to build 3D devices on planar substrates [1]. However, the fabrication of high-resolution monolithic macroscopic 3D structures of arbitrary ‘freeform’ stereometry necessitates the deployment of alternative concepts. In recent decades, state-of-the-art maskless techniques [2] have emerged, including stereolithographic methods, additive manufacturing, formative casting and molding. Direct laser materials growth and processing [3] now incorporates laser-induced material transfer [4], even of sensitive biological materials [5], and artificial intelligence optimization methods [6].

Additive manufacturing by laser sintering methods is now an industrial reality, achieving, however, a relatively low spatial resolution on the scale of tens of micrometers. Multiphoton laser polymerization [7] achieves 3D structures with submicron precision. Experimental results have shown the capacity for a sub-100 nm resolution by using special polymer and hybrid resins [8]. Even though the latter method is capable of high-resolution fabrication of predesigned freeform 3D objects, it exhibits significant disadvantages of limited physical processing volume, low processing speed, high complexity and high cost.

Self-assembly methods have been developed for the formation of 2D and 3D lattices using polymers [9], liquid crystals, nanocrystals and polystyrene microspheres [10]. These lattices have been used in turn as templates for lithographic processing, assuming the role of exposure masks in 2D and 3D device fabrication, which are both of interest in nanophotonics [11]. This method enables the formation of regular lattices, but it is not capable of freeform 3D object fabrication. The complexity, the reliability and fabrication speed usually associated with multistep processing are persistent disadvantages.

Nature offers an unlimited variety of 3D structures, organs, optimized via biological evolution to perform extremely complex physical, chemical and biological operations [12]. Organs serve the specific needs of living creatures and plants, while exhibiting unparalleled curves and forms [13]. The interaction of man with the natural world has been the inspirational playground for the conceptualization and fabrication of artificial functional objects since the ancient times [14]. This art evolved to the modern, broad and rapidly expanding technological domain of *biomimicry* and *biomimetics* [15]. Advanced concepts and devices mimicking the forms, functions and processes of plants and animal life cover several diverse fields and respective technologies, from biological chemistry [16] to mechanics [17], photonics [18] and machine vision [19], to information technology and neuromorphic processors [20], fluid dynamics [21] and others that have recently been reviewed [22,23].

Focusing on photonics, considerable research has been directed toward the intriguing phenomena of structural coloration and the compound vision of insects. Nanostructures of butterfly wings [24] and scarab bodies [25] have been thoroughly studied, being most relevant to diffractive and photonic band gap (PBG) structures [26]. The fabrication of 3D PBGs in the visible spectrum is still presenting several technological challenges [27]. Natural structures, such as those seen in the *Morpho rhetenor* butterfly [28], cicada wings [29] and golden beetles [30], could inspire novel solutions [31].

More relevant to the paradigms addressed in this work is the compound eye of insects. Following the classical studies of Hooke [32], this vision organ has attracted huge interest and inspired the technology of microlens arrays. Extensive studies have been devoted to its anatomy and biological operation [33], as well as its potential in biomimetic applications [34], from imaging [35] and wide-angle machine vision [36], to sensing [37] and micro/nanofabrication. A variety of methods have been deployed for the fabrication of artificial biomimetic lens arrays [38]. Depending on the materials used and the dimensional scale, combined subtractive and additive methods have been employed [39], including lithographic processing [40], femtosecond laser microfabrication [41,42,43], self-assembly and template replication [44,45] and additive manufacturing [46]. Of relevance is the formation of microlens arrays distributed over a hemispherical surface [47,48], closely mimicking the natural compound eye. Ultrawide hybrid optics for machine vision [49] have also been demonstrated, enabling superposition [50] or apposition [51] imaging for enhanced image quality [52] and wide-field angular coverage.

Natural microneedles found on the bodies and wings of insects represent another example of interest in this work. Biomimetic microneedle array technology [53] is attracting considerable attention, finding applications in transdermal drug delivery [54], the administration of immunobiological substances, cosmetic treatments, cell sensing and disease diagnosis [55]. The fabrication of such structures has been achieved by a variety of lithographic microfabrication methods [56], including multiphoton polymerization [57], and etching [58]. Photonic nanotips have been developed in silicon for antireflection coatings (black silicon) [59], fiber optic nanotips for scanning near-field optical microscopy (SNOM) [60], fiber sensors [61], photoimprinting [62], metamaterial nanotip [63], silicon carbide photonic arrays for light delivery [64], and spectral control [65]. On the other hand, and beyond bioinspiration, the realistic biomimetic nanotip arrays such as, for example, the semiconductor nanotips inspired by cicada wings [66] have been overlooked and they merit further attention.

Direct replication of natural elements by casting has been developed as a supporting step in scanning electron microscopy (SEM) of plant specimens [67,68,69] following procedures like those applied in dentistry. More recently, tin-oxide calcination of *Morpho* butterfly wings [70] resulted in biomimetic photonic structures and demonstrated variable structural coloration. Direct biomimetic replication of butterfly wings has also been performed by soft lithography to investigate optofluidic synergetic properties [71]. In our recent research [72], we have replicated artificial diffractive elements in aerogels and demonstrated stepwise systolic processing, yielding physically downsized micro- and nanoscale structures. The most recent replication of compound eye of dragon fly [73] has demonstrated gradient ommatidium arrays for multi-focus imaging and enhanced vision acuity. In addition, cicada wings have been used as templates for fabricating flexible polydimethylsiloxane (PDMS) nanoporous substrates, which have been loaded with uniform Ag nanoparticles and employed in portable surface-enhanced Raman scattering (SERS) systems [74].

Relevant to our work is the thermal transformation of silica aerogel into dense fused-silica glass [75,76]. Vitrification of sol–gel casts has been used to form microlens arrays [77]. In our previous studies [78], we laser patterned aerogel solid objects to produce surface relief and in-volume-void embedded microstructures. Subsequent thermal processing of these decorated monoliths produced miniature fused silica replicas of the original master objects. Both the bulk monolith and its embodied micropatterns were conformally, simultaneously, and analogously downsized. This *systolic transformation* enables the manufacture of 3D freeform solids that are geometrically affine to aerogel master (parent) objects but have dimensions which are submultiple to their master. As a result, the smallest feature crafted by any pattering method is transformed to its affine clone feature, size-reduced by a systolic factor (SF) determined by the materials and methods used. Consequently, this method facilitates *super-resolution* fabrication of functional 3D devices of arbitrary, freeform stereometry, that is not achievable by any other means to date.

In the present work, we replicate *Bioarchitectural* forms in aerogel and transform them into functional elements. These elements utilize the organized and unified structure of the specific bioarchitectural design, without necessarily imitating or reproducing the biological processes of the master (parent) organ. Essentially, they are a kind of *Bioarchitectonic* elements. Despite being produced through this relevant approach, they differ from the established modern concept of *Biomimetics*.

Focusing on optics and photonics, we replicate bioarchitectures in ultra-porous aerogels [79]. We form, transform, demonstrate, and characterize the functional operation of these 3D artificial photonic elements for the first time to our knowledge. Although silica aerogels are ultra-porous materials exhibiting extremely low mass density (<0.25 gcm^−3^) and refractive index (n < 1.1), our results prove that they can be used to form ultralightweight refractive optical elements. In our approach, we have chosen the well-known paradigm of compound eye and investigated three affine versions of the complete head of European hornet *Vespa crabro flavofasciata* embodying *ommatidia* microlens arrays. In another paradigm, the ‘golden beetle’ scarab *Protaetia cuprea phoebe* provided *microtrichia* arrays found on its wings. They have been transformed into optically transparent microneedles that waveguide light and deliver it through their nanotips. The photonic performance of microtrichia replicas is completely unrelated to the biological function of natural microtrichia, thus highlighting the distinct *bioarchitectonic* approach we address in this work.

## 2. Materials and Methods

### 2.1. Instrumentation

A scanning electron microscope (SEM) Zeiss EVO MA10 (Carl Zeiss AG, Oberkochen, Germany), equipped with LAB6 electron source, was used for the analysis of both the natural and the artificial objects.

Optical imaging was performed by use of OPTIStar OS-30T microscope in conjunction with a CMOS camera C-B5 (OPTIKA S.r.l, Ponteranica, Italy). A beam profiler SP620U (OPHIR-Spiricon LLC, North Logan, UT, USA) was also used for image analysis. Special optical arrangements were configured to manage illumination and RGB color filtering (Edmund Optics, Barrington, NJ, USA).

Aerogel and xerogel syntheses were performed using in-house clean-room chemical processing facilities. Replication templates were cast from natural wax and polymer compositions to cope with the sensitivity of biological structures. Standard commercial materials have also been used including PDMS SYLGARD 184 (Dow Corning GmbH, Wiesbaden, Germany) and UV-curable resin ORMOSTAMP^®^ (micro resist technology GmbH, Berlin, Germany), along with proprietary polymer blends developed in-house.

Supercritical drying was performed using high temperature, high pressure autoclaves having volumes of 75 mL and 1000 mL (Parr instruments, Moline, IL, USA). Thermal processing and systolic transformation of solids were performed by use of computer controlled high temperature ovens (Thermanys S.A., Thessaloniki, Greece).

### 2.2. Elements of Insect Organs

Experimentation with parts of insect bodies does not raise bioethics concerns [80] and thus we selected and used specimens of deceased insects. In the first paradigm of this work, we focused our interest on the compound eyes of hornets [81] and the *microtrichia* arrays of scarabs [17]. Both species are found in abundance in Greece and the Mediterranean basin. The European hornet *Vespa Crabro Flavofasciata* provided its microlens ‘*ommatidia*’, while ‘the golden beetle’ scarab *Protaetia Cuprea Phoebe*, provided its microneedle ‘*microtrichia*’ arrays.

#### 2.2.1. Ommatidia

The remarkable vision capabilities of fast–flying insects, and other arthropods, are attributed to their compound vision system. SEM imaging was used to analyze the natural compound eye of *Vespa crabro flavofasciata,* as shown in Figure 1. The compound eye consists of *ommatidia* arranged radially in a close-packed hexagonal lattice. Figure 1a depicts the frontal view of hornet’s head, while Figure 1b presents a close-up view of *ommatidia.* Each *ommatidium* comprises an outer part, the *cornea*, which is the hexagonal plano-convex microlens objective, although pentagonal and squared elements are also found. The diagonal dimension of the hexagon measures as D ~ 30–40 μm. According to biological studies [81], the cornea’s focal length is about f ~ 110 μm. The finite conjugate ratio imaging approach, however, yields an overestimated value. Furthermore, analysis of SEM images of refs. [33,81] suggests that the radius of curvature of the cornea is R ~ 70 μm and its thickness is approximately t ~ 100–120 μm. Considering that this thick plano-convex element provides the focusing action, its paraxial focal length, f, can be approximated by:(1)$$\frac{1}{f}=\frac{n-1}{R}$$
where R is its radius of curvature and n its refractive index. The focal length value f ~ 110 μm implies a cornea refractive index n_cornea_ ~ 1.63, which contradicts the expected refractive index of chitin, n_chitin_ ~ 1.525. This discrepancy can be corroborated by considering the effective refractive index of cornea’s multilayer structure. Such a multilayer structure resolved in the SEM images in ref. [33] and ref. [82] is also observed in our SEM analysis.

The accuracy of the dimensional measurements in SEM and optical microscopy depends on factors such as the magnification, the instrument performance, the materials nature, as well as the human error. In this work, we used the dimensional tools provided by the SEM, which deduced an experimental error of σ_SEM_ = ± 2 nm at the highest magnification (×12.11 k) applied. We note, however, that the said dimensional accuracy depends on the image quality achieved. Consequently, it does not apply to low quality imaging such as those that are blurred or affected by electron charging. Optical analysis exhibits an experimental error of σ_optical_ = ± 1 μm. In addition, we also strongly underline the natural variance in geometry and dimensions among individual ommatidia, as well as their dependence on specific location at the vision organ. We also note that a number of lenslets in specific areas of the eye are pentagonal and squared, conformally arranged to fill the close-packed microlens array.

Anatomically, each primary hexagonal cornea element collects light emanating from the object space within a narrow field of view and transmits it along its optical axis. A secondary conical transparent element, about 60 μm long, receives the light and couples it to the underlying transparent fiber tract, known as the *rhabdom*, which has a core diameter of about 2 μm and length of approximately 370 μm. Each rhabdom is enclosed all along its length in a low refractive index iris cell, forming an optical waveguide. Technically, this is equivalent to a microlens coupling light to an optical fiber via a conical taper section. The light collected and waveguided reaches the underlying photoreceptor cells and synthesizes a relatively low-resolution mosaic image. This is the case of apposition imaging where image segments generate individual signals. 

#### 2.2.2. Microtrichia

Natural structures of ordered *microtrichia* ensembles are found on the hard and soft parts of the wings of scarabs and other insect species. These quasi-periodic microneedle arrays form nearly perfect hexagonal lattices over large surface areas. They serve various actions [83,84], including antibacterial protection, airflow control for enhanced aerodynamics, environmental sensing, body-temperature control, and prevention of body dehydration.

Members of the *Scarabaeidae* family, and generally the coleoptera, have two pairs of wings, the forewings, known as *elytra*, and the hindwings. Figure 2 presents SEM micrographs of microtrichia on the body of a *Protaetia Cuprea Phoebe* specimen. Figure 2a,b depict representative samples of the forewing *elytron*, which is the usually stiff member covering either completely, or partially, the abdomen. Both surfaces of the *hindwings* and the ventral side of the *elytra* are covered with *microtrichia*. Figure 2c,d present SEM micrographs of hindwing *microtrichia*. Each *microtrichium* is a conical element having an average height of about 10–20 μm and a base diameter of approximately 5 μm. The exact shape and dimensions of microtrichia vary depending on their specific location, as illustrated in the representative ensembles of Figure 2. Based on the SEM images, we estimate the apex radius of curvature of a microtrichium to be in the range of 100 ± 20 nm. This estimation represents the maximum value considering the limitation of instrument’s resolution and the focusing uncertainties imposed by the high curvature of the sample, together with the local surface conductivity, and the mechanical and thermal stability of the imaged tip under electron irradiation.

In our experimental work, we selected and categorized natural specimens. After thoroughly cleaning them in ethanol under sonication for 20 min, the specimens were stored in methanol and prepared for further experimentation.

### 2.3. Aerogel and Xerogel Solid Replicas

This section addresses the replication of insect body elements to form their 3D ‘clones’ made of silica aerogel. Silica aerogels [85] are ultra-porous solid foams made of an air-filled solid skeleton. These ultralight solids exhibit over 95% porosity, a mass density, usually less than ρ < 0.2 gcm^−3^, and a correspondingly low refractive index, typically n < 1.1.

The fabrication of 3D nano-sculptured aerogel replicas involved the development of soft-lithographic methods using natural materials to preserve the structural integrity of the specimens. Negative 3D master casts have been formed by using special wax achieving remarkable fidelity. Two additional replication steps followed to accurately clone the bioachitectures in aerogels, using polymer blends, PDMS and ORMOSTAMP^®^ resins for the final replication stages.

Sol–gel synthesis was applied for the fabrication of aerogels using tetramethoxysilane (TMOS) as the silica source, yielding highly uniform fully transparent glassy networks suitable for optical applications. Our synthesis commenced by mixing TMOS with methanol (MeOH) at a molar ratio of 1:12 M. A solution of deionized water and the basic catalyst ammonium fluoride (NH_4_F) at 4:3.5 × 10^−3^ M molar ratio was added to the silica sol to initiate chemical condensation reactions yielding silica alcogel. The silica alcogel was poured into negative templates and left to condensate. After ~20 min gelation time the samples were transferred for aging in TMOS/MeOH. The aging process strengthened the porous silica network and increased its specific surface area. The wet gel replication of the natural object form was successfully completed, and the replica was delivered for drying, thus forming the solid aerogel object.

High temperature supercritical drying (HTSCD) of the wet alcogels produced highly porous aerogel monoliths of large size. This technology has been successfully used in visual arts [86], and the present work is a proposition for bioarchitectural design. Appropriate control of the supercritical conditions equalizes the pressure and prevents shrinkage and collapse of the solid network. In addition, we implemented a more facile ambient pressure drying (APD) method [87] involving a series of solvent-exchange cycles for effective chemical modification of the skeleton. In our experiments, the chemically treated gel was allowed to dry naturally at room temperature for 12 h, followed by heat treatment at 150 °C to ensure the complete removal of the residual solvents.

Natural environmental drying (NED) is a third method, applied to untreated wet gels under ambient environmental conditions, yielding silica xerogels. Experimentally, we have applied slightly elevated temperatures ~60 °C under atmospheric pressure to accelerate the drying process. Microscopically, capillary forces induce non-uniform stress on the pore walls during liquid extraction. As a result, the network collapses into the bulk volume formerly occupied by the dispersion liquid. The gel shrinks, and a xerogel solid having lower porosity in the range of ~10–70% is formed. The level of porosity can be controlled upon synthesis. This downsizing effect may be considered as the first stage of the *conformal systolic miniaturization* to be followed by the subsequent stages implemented in the course of this work. We note that shrinkage is generally absent in aerogel formation, owing to the equalization of capillary tension forces in the porous structure. However, a minor degree of dimensional reduction may occur in practice due to experimental and physical uncertainties involved in high-temperature supercritical drying.

### 2.4. Systolic Miniaturization of Aerogel and Xerogel Replicas

The advent of monolithic aerogel [75] has been followed by the breakthrough transformation of the aerogel monolith into high-quality silica glass, via thermal sintering and viscous flow vitrification [76,88]. The versatility of sol–gel chemistry offers substantial advantages, allowing the formation of complex glass compositions, doping and structural modifications using relatively low-temperature, environmentally sustainable processing.

Conformal systolic transformation has been extended to include both artificial holographic and natural pattern miniaturization [79]. This method allows a small 3D object to shrink into its exact clone having reduced dimensions. Systolic miniaturization effectively downsizes the minimum resolvable features of an object, producing its replica with super-resolution.

In the present work we utilized systolic processing to create functional bio-architectonic elements for photonics. The degree of miniaturization is quantified in terms of the linear systolic factor SF× defined as:(2)$$SF\times =\frac{L_{i}}{L_{f}}={\left(\frac{\rho_{f}}{\rho_{i}}\right)}^{1/3}$$
where L_i_ and L_f_ are respectively the initial (before systolic processing) and the final (after systolic processing) linear dimensions, and ρ_f_ and ρ_i_ are the respective final and initial mass densities of the solid objects. We note that the ultimate density to be reached is that of dense fused-silica, being ρ_fs_ ~ 2.2 gcm^−3^.

Experimentally, thermal processing commences at ~250 °C, progressing to 500 °C to oxidize the organic content, and reaching ~900 °C for diffusional sintering. The final vitrification stage achieved by viscous flow occurs in the range of ~1100–1250 °C, yielding fused silica through the total collapse of the porous skeleton. Systolic processing in xerogels was also performed by following similar procedures. In this case, ambient gel drying produced an initial downsizing effect of SF × ~2.2 at 60 °C under atmospheric pressure. Slightly higher systolic factors, by about 10–20 %, have been observed for surface relief features, which may be attributed to increased surface tension effects. Controlled stepwise systolic transformation of complex object geometries, using discrete process stages, was successfully demonstrated for the closed surface topology. This multiplicative downsizing effect yields controlled systolic factors ranging from SF × 1.5 to 5.0, depending on the process protocols. The resulting fused silica object is a miniature replica, ‘clone’, of the original nanoporous solid monolith, which is fabricated with feature resolution SF × higher than the feature resolution of the original object. Ongoing optimization efforts aim to maximize the systolic factor and replication fidelity, with prime focus particularly on optical quality materials for nanophotonic devices.

## 3. Results and Discussion

### 3.1. Bioarchitectonics in Biomimetics

*Biomimetics*, a concept and technology developed over thousands of years, has evolved into a broad, rapidly expanding, and highly interdisciplinary field spanning many disciplines. In its modern definition, biomimetics involves the study of biological forms, actions, mechanisms, and processes, targeting to synthesize artificial products and mechanisms that mimic the natural ones, from the macroscopic to the molecular scale. Current developments embrace micro/nanotechnology methods, yielding novel chemical/biochemical processes and cutting-edge functional devices, some of which are already reaching the marketplace.

In a broader context, *Biomimetics* would be conceived as a subset notion of *Bioinspiration*. This encompasses all concepts, forms, topologies, mechanisms, methods, and products that have a reference to biological development. However, one may note that the majority of biomimetics research and technology concerns mimicking biological forms, mechanisms, and processes. 

Building on ideas of architecture and art, we explore in the present work the independent use of forms that do not necessarily mimic the functions of their biological generators. This *architectonic* approach represents a new and emerging category within the broader biomimetics sector, distinct from current mainstream biomimetics. By a fruitful selection of structural forms, materials and methods, we create *Bioarchitectonic* devices that function differently than their biological counterparts. 

In this context, we examined two well-known paradigms, the compound eye *‘ommaditia*’ and the microneedles *‘microtrichia’*. We fabricated aerogel and xerogel devices of compound eyes of hornet, of the species *Vespa Crabro Flavofasciata* and *microtrichia* ensembles of scarabs, of the species *Protaetia Cuprea Phoebe,* as functional photonic devices. In spite of their extremely low mass density and refractive index, the optical performance of these devices demonstrates their potential for manufacturing ultra-lightweight optical elements. In addition, systolic miniaturization of these replicas resulted in transparent fused silica glass photonic devices, which are miniaturized clones that perform differently from their natural parents.

### 3.2. The Paradigm of Biorchitectonic Compound Eye

Silica aerogel replicas of the compound eye of the hornet *Vespa crabro flavofasciata* have been fabricated using high-temperature supercritical drying and multistep ambient-pressure drying of wet gels, as described in previous sections.

Figure 3 presents an example of an aerogel replica monolith fabricated via high-temperature, high-pressure supercritical drying. Figure 3a presents a far view of the hornet’s head replica, while Figure 3b provides a close-up view of the hexagonal microlens array structure selected from the indicated spatial location. Although the 3D aerogel object consists of a very brittle nanoporous air-filled skeleton, these replicas exhibit remarkable structural uniformity and reproduction fidelity. The average diagonal dimension of each lenslet ranges from D ~ 30–35 μm, indicating a ~ 10% deviation from the dimensions of the natural cornea. We reiterate that the natural variance and the plurality of microlenses prevent exact one-to-one comparison between the fabricated elements and their natural counterparts.

The optical function of the fabricated compound eye optic was investigated by using the optical microscope under white-light, and color-filtered, plane-wave transillumination, directed bottom-up through the microscope column. This arrangement allowed for the observation of the refractive surface and the ensemble of focused beamlets, the latter serving as the experimental functional test case. A 20× objective having numerical aperture NA = 0.2 was used for imaging, in conjunction with the CMOS camera. In addition, the imaging beam profiler operating at low light levels and below saturation allowed for accurate intensity quantification, as well as facile visualization by employing pseudo-coloration. Figure 3c shows the image of the surface captured by the beam profiler. The surface was set at the working distance of the microscope objective, as indicated in the drawing of Figure 3d. The microlens array foci, as observed in Figure 3e, are recorded by lowering the microscope sample-stage as illustrated in the drawing of Figure 3f.

Experimentally, quasi-Gaussian focal spot profiles were observed at a distance h ~ 195 ± 5 μm above the aerogel surface, as measured by the micrometric translation stage of the microscope. The measured FWHM can be related to the ideal Gaussian spot size by:(3)$$FWHM=1.18\times w_{o}=\frac{2\times 1.18\times \lambda }{\pi }F\#$$
where w_o_ is the ideal Gaussian spot size, λ is the wavelength of light and F/# = f/D is the f-number of the focusing optic. The elevation h ~ 195 μm and wavelength λ = 500 nm, imply f-number of F/6, which yields an estimated FWHM ~ 2.2 μm. This result is in contrast to the measured FWHM ~ 15 ± 1 μm value shown in Figure 3e. Therefore, we may reasonably conclude that the measured apex-to-focal point distance, h, corresponds to the ‘back focal length’, f_b_ ~ 195 μm of the thick optical element.

We further investigated by SEM microscopy to directly measure the radius of curvature of both the natural cornea specimen and its aerogel replica, as shown in Figure 4. Owing to the fragility of aerogel, a precise dissection of the aerogel replica was not possible and thus we employed a high inclination profile to verify the result.

Figure 4a presents the cross section of the natural compound eye of a hornet specimen, indicating the cornea surface (A) and the underlying multilayer structure (B). This is followed by the conical taper segments and rhabdoms (C), the latter being damaged due the dissection. Notably, fine damages and blemishes on the natural eye are replicated in the artificial elements.

Figure 4b shows a close-up view in which we measured the linear dimensions of the chord, c ~ 33.75 ± 0.01 μm, and the sag, s ~ 2.85 ± 0.01 μm, which were used to estimate the radius of curvature.

Assuming a spherical surface of the cornea, we applied the power-of-point theorem for the major circle of the sphere:(4)$${\left(\frac{c}{2}\right)}^{2}=s\left(2R-s\right)$$
and estimated the radius of curvature as R ~ 51.38 ± 0.02 μm. Bearing in mind the dimensional tolerances and the natural variance of the *ommatidia* geometry, this value agrees well with the value estimated using projection geometry applied on the high inclination profile of the aerogel replica shown in Figure 4c.

We used nitrogen porosimetry and volume-fraction estimation (SiO_2_-Air) to determine the effective refractive index n_aero_ of the high purity and optical quality aerogel. These samples exhibit ~90% porosity, yielding mass density of ρ_aero_ ~ 0.22 gcm^−3^, and n_aero_ ~ 1.046 at λ = 500 nm.

Considering that only the convex surface of the aerogel replica, having radius R ~ 47 μm, is involved in the focusing action, we use Equation (1) and estimate the paraxial focal length of the aerogel microlens as f_aero_ ~ 1030 μm, and the f-number, as F / 32. Using this value in Equation (3) we estimate the FWHM ~ 12 μm, which is in reasonable agreement with the experimentally measured FWHM ~ 15 ± 1 μm shown in Figure 3e.

The free-space propagation of a real, non-diffraction-limited, optical beam can be described in terms of the beam propagation factor M^2^ [89] defined as:(5)$${Μ}^{2}=\frac{W_{ο}\Theta_{ο}}{w_{o}\theta_{ο}}=\frac{\pi }{\lambda }W_{ο}\Theta_{ο}=\frac{\pi }{\lambda }\frac{D}{2f_{b}}\frac{FWHM}{1.18}$$
where, λ, is the wavelength of light, W_o_, the real focal spot-size and, Θ_ο_, the measured divergence. These values differ from those of an ideal diffraction-limited Gaussian beam, having respective spot-size, w_o_, and divergence, θ_ο_. In the present case, we derived the divergence, Θ_ο_, by considering the filled microlens aperture, D, the focal plane position along the propagation axis measured as the back-focal-length, f_b_, and the focal spot W_o_, determined by the measured FWHM at the focus. Applying Equation (5) with these experimental parameters we estimate M^2^ ~ 7, implying a considerable deviation from an ideal Gaussian.

This discussion highlights the potential of ultra-porous aerogel materials to produce refractive optical elements, provided that high optical quality is attained. Unlike microlens arrays produced by other methods, these elements are extremely lightweight, with an estimated mass of the individual refractive lenslet to be of the order of ~1 ng. The near unity refractive index naturally produces a relatively long focal length, even though it can be tuned by adjusting the materials density and the micro-optic curvature. Furthermore, the extremely low thermal conductivity of aerogel makes it attractive in various applications.

Similar results of optical function have been obtained using quasi monochromatic illumination provided by the blue (B) and red (R) dichroic filters of the RGB standard, respectively centered at 470 nm and 630 nm of the spectrum. Figure 5b shows selected areas of the compound optic surface relayed using the blue filter (B) and Figure 5c presents the corresponding microlens foci. The curvature of the compound eye surface prevents simultaneous sharp image focus across the entire image field. The focusing conditions are presented in the schematic of Figure 5a with exaggerated dimensions for clarity. The paraxial ray propagates through the thick bulk and focuses at the back focal points, F_B_, and F_R_ for the blue and red respectively. These focal points are positioned respectively at the back focal lengths f_b_(B) and f_b_(R) distances, as they are measured from the microlens apex. Considering the focal points F_B_ and F_R,_ and the second principal point H_2_ defined on axis by the principal plane H_2_, the focal lengths of the optic are f_B_ = |F**_B_**H_2_| and f_R_ = |F**_R_**H_2_|, for blue and red light respectively.

Experimentally, the elevation distance between the two image settings, i.e., focus on surface and focus on focal plane, records the back-focal length f_b_(B) ~ 185 ± 5 μm of the individual lenslet for blue light as shown in Figure 5b,c, and the back focal length f_b_(R) ~ 200 ± 5 μm for red light as depicted in Figure 5d,e. Assuming a common achromatic second principal plane H_2_ we can make an account of the aerogel dispersion, v, as:(6)$$v=\frac{1-n_{aero,R}}{1-n_{aero,B}}=\frac{f_{B}}{f_{R}}\sim 0.98$$

We note that this value is not the Abbe V-Number but it is only a relevant figure-of-merit.

In the second stage of our investigation, xerogel structures were fabricated by casting and drying alcogels. Particular attention was paid to ensure uniform solvent extraction and smooth demolding to prevent dimensional discrepancies. Natural drying is the primary systolic step, delivering a downsized monolithic xerogel replica of the hornet head as shown in Figure 6a. Figure 6b presents a close-up view of the hexagonal microlenses. By averaging the measured hexagon diagonals as D_xero_ ~ 16 ± 2 μm we estimate a systolic factor SF × ~2.1. Figure 6c,d, present the two focusing positions at the surface and the microlens focal plane, respectively. The focus was positioned approximately at 90 ± 5 μm above the object surface. A Gaussian focal spot having FWHM ~ 4 μm is observed in Figure 6d. After the primary systolic process, the resulting microlens radius of curvature is estimated as R_xero_ ~ 22 μm. In addition, nitrogen porosimetry of xerogel gives a porosity value of ~50 %, implying a xerogel refractive index n_xero_ ~ 1.23. These values yield an estimate of the focal length f_xero_ ~ 98 μm, at F/6. By using Equation (3) we estimate the diffraction-limited Gaussian focal spot FWHM ~ 2.2μm that is comparable to the observed FWHM ~ 4 ± 1 μm in Figure 6d. Using Equation (5), we estimate the factor M^2^ ~ 1.8 verifying the nearly diffraction-limited performance of the microlens.

Figure 7 illustrates the imaging function of the xerogel microlens array under quasi monochromatic conditions provided by the blue (B) and red (R) filters of the standard RBG filter set. We reiterate that the high curvature does not allow sharp imaging over the field. A considerable difference between the foci positions measured experimentally is observed. The difference in focus setting in Figure 7a,b measures the back focal length f_b_(B) ~ 60 ± 5 μm for the blue range. Similarly, using Figure 7c,d, we measure the back focal length f_b_(R) ~ 90 ± 5 μm for the red spectral region. Applying similar arguments, we infer by Equation (6) a measure for the xerogel dispersion figure-of-merit v ~ 0.73.

Viscous flow vitrification of both aerogel and xerogel replicas was performed through thermal processing at approximately 1100 °C. Figure 8a shows a miniaturized hornet head replica composed of dense fused silica, fabricated by vitrification of a monolithic aerogel replica. Submicron original defects and droplet-type defects, probably formed upon solidification of xerogel microparticles. By comparing the diagonal of the original aerogel master object to the measured diagonal of the individual microlens elements in Figure 8b, D ~ 10 μm, we deduce the systolic transformation factor SF × ~3.4.

The optical image of the surface under white light illumination is depicted in Figure 8c. The focusing action of the microlenses in Figure 8d allows for the direct measurement of the microlens back focal length, which is approximately f_b,f-s_ ~ 30 μm. Considering the systolic factor SF × 3.4, we can estimate the radius of curvature of the individual microlens element to be R_f-s_ ~ 14 μm. Such a high curvature is comparable only to self-assembled microsphere cast arrays [90], or nanofabricated planar arrays [91]. Recent approaches aim to provide microlens arrays spherically distributed to cover an ultrawide field of view [92]. The significant difference between these biomimetic approaches and our results is that the bioarchitectonic array covers conformally an oval surface of very complex topography, as it is endowed by nature. In fact, the radius of curvature of this surface varies significantly along the axes of vision to provide the best possible field coverage and the vision processing required by the insect. The contour schematic of Figure 8e presents anatomical details and the respective radii of curvature of the different parts, estimated here by SEM imaging and projection in the range of approximately 70 μm to 1068 μm. This clearly represents a step beyond the current state of the art.

Using the refractive index value of fused silica n_sil_ = 1.46 at λ = 500 nm, we estimate the microlens focal length to be about f_f-s_ ~ 35 μm, at F/3.5 and FWHM = 1.32 μm for the diffraction-limited Gaussian. A polychromatic focal spot of FWHM ~ 3.5 ± 1 μm is observed in Figure 8d. The deduced propagation factor M^2^ ~ 3 indicates a deviation from the diffraction-limited performance, which could be due to refractive index and shape errors in the densified silica solid.

### 3.3. The Paradigm of Bioarchitectonic Microneedles

In this section we demonstrate the replication of microtrichia arrays, a representative case in which the fabricated elements serve a function totally independent of their original biological counterparts.

Microtrichia cover the forewings, *elytra*, and the hindwings of the scarab. To explore the optical properties, we employed APD processing to fabricate aerogel replicas of microtrichia found on different locations of the wings and body of the insect.

Figure 9 shows the monolithic aerogel replicas of elytra microtrichia. These replicas were fabricated from different areas of the specimen, with an example depicted in Figure 9a and its detail in Figure 9b. The replicas are conical elements of an approximate height of 5 μm and a cone base diameter of about 2–3 μm. The apex radius of these needle-like structures is around 100 nm, indicating a slight shrinkage of approximately 20 %, attributed to the spring-back effect inherent to the APD process. Figure 9c,d demonstrate the optical functionality of the aerogel microneedles under white light plane-wave transillumination. Figure 9c presents an image focused on the cone base, while Figure 9d shows the light emitted from the nanotips as it is detected at the resolution limits of the microscope. Light is guided through the transparent microneedle’s trunk and is decoupled producing the beamlet spots observed at the nanotip.

The monolithic aerogel replicas of hindwing microneedles from three different locations have also been investigated and are presented in Figure 10a–c. In this case, smaller structures are observed with an average height of 2–4 μm and a cone base diameter of ~2 μm or smaller. Figure 10d demonstrates their optical performance. The noticeable variation in surface curvature results in differing focusing conditions. Areas of interest with the image focus at the microneedle’s cone base are encircled with solid line, while areas with image focus at the nanotip are encircled with a dotted line. We underline the color change at the center of the light circle, indicating a region of high intensity resolved at the limits of detection. The reproduction fidelity at this miniaturization level is particularly noteworthy.

Further elaboration of the method addressed the xerogel replication of both elytron and hindwings. Figure 11a,b present the replication results of two different areas of the elytron, corresponding to the specimens shown in Figure 2a and 2b respectively. The radii of the microneedle apex are in the sub-100 nm range, with SEM images clearly reaching the resolution limits of the instrument. We note that due to experimental complexity, it is practically impossible to identify the exact one-to-one correspondence between the natural microtrichia and the replicas produced. However, the elytron area and associated replicas are distinguishable, allowing us to estimate an average systolic factor in the range of SF × 2.5. Figure 11c,d illustrate the optical performance of xerogel replica the elytron. Examples of focusing on the base of the conical microneedles are encircled with solid line. Focal spots of the microneedle apex are encircled with dashed lines. The apparent focal spot, observed by the microscope is in the ~1 μm range, clearly reaching the resolution limits of the instrument.

Examples of hindwing replicas in xerogel are also presented in Figure 12. Figure 12a–c are SEM micrographs of replicas from three different areas of the hindwings. The dimensions of the base of the conical microneedles vary depending on the location, ranging from approximately 1 μm to 2 μm, with heights around 2 μm to 3 μm. In Figure 12d the image focus is at the cone base, while Figure 12e shows the corresponding focal spots at the nanotips, which are about 1.5 μm in size and reach the diffraction limits of the instrument.

In the next task, we applied the final systolic step in both aerogel and xerogel replicas. Thermal processing in the range of 1100 °C resulted in a total dimensional reduction of approximately SF × 4 and the fabrication of fused silica clones. Examples of vitrified xerogel microtrichia replicas are depicted in Figure 13. Figure 13a,b present fused silica replicas drawn from two different areas of the hindwing. The cone base diameter of the observed nanoneedles is about 1 μm, with the submicron cone trunk extending to nanotip apex radii estimated in the sub-50 nm range. Increasing SEM magnification produces image blurring, making the nanotip unresolvable. This blurring is likely due to inhomogeneous conductivity at the tip, charging and instabilities induced by electron beam irradiation. Figure 13c shows the paradigmatic light guiding and focusing functions of the replicated nanoneedles. When the image focus is at the surface, the cone bases are encircled in a solid line. Focusing a few microns above the surface, emission from the nanotips, with spots approximately 1 μm in size is observed, as indicated by the dotted line, clearly reaching the limits of detection. 

Considering artificial nanotip probes, such as fiber tips used in SNOM or photon scanning tunneling microscope (PSTM) we note that the dimensions of the nanotips fabricated above have apex radii of similar size, and thus can provide the like operations and potentially multiple sensing applications in the nanoscale. Specifically, a nanotip array in nanometric proximity to a back-illuminated, totally reflecting surface can frustrate the total reflection and retrieve light from the evanescent field, thus enabling the detection of materials or agents on the sample surface. On the other hand, a significant portion of light waveguided in the nanoneedle reaches the propagation cut-off due to the nanotip dimensions and is backreflected.

Approaching the nanotip on the surface frustrates total reflection and couples light into the sample. Furthermore, these nanotips can be transformed into plasmonic arrays by deposition of noble metal (Au, Ag, etc.) nanolayers, which will extend their interactions beyond the purely dielectric domain. Conclusively, the complex stereo-topography of the fabricated bioarchitectonic nanotip arrays can offer several advantages in sensing and light-matter interactions, and further work is in progress.

## 4. Conclusions

Biological development offers an unlimited variety of forms, exhibiting both unsurpassable complexity and mathematical order. This work draws inspiration from architectural concepts that have been utilized for millennia and explores the application of bioarchitectural forms and their derivatives to create 3D functional devices for photonics. These devices do not necessarily replicate the mechanisms or processes of the parent organisms, instead they offer fundamentally different functionalities that may not be achievable through other means. 

In this work, we investigated a novel bioarchitectonic 3D nanofabrication approach based on the direct replication of structural elements of insects using ultra-porous silica aerogels. Aiming at photonics devices and applications, two paradigms have been investigated using natural specimens of the European hornet *Vespa crabro flavofasciata* and the ‘golden beetle’ scarabeo *Protaetia cuprea phoebe*. Aerogel and xerogel replicas of hexagonal *ommatidia* cornea of the compound eye and *microtrichia*, microneedles, were successfully fabricated, and their functionalities were demonstrated and analyzed. First results confirmed that silica aerogels having extremely low refractive index can be used to produce ultralightweight refractive micro- and nano-optics. In the following stages, these freeform bioarchitectonic elements underwent systolic transformation, which resulted in their densification and dimensional miniaturization. This process preserved their conformality to the original master structures provided by the parent biological organ. Three-dimensional fused silica clones of the natural compound eye having hexagonal microlenses with approximately 10 μm diagonal, focus collimated visible light at focal lengths f ~ 35 μm with a f-number, f/3.5. Transparent fused silica replicas of microtrichia waveguide and deliver the light through their sub-50 nm nanotips.

Such bioarchitectonic concepts and methods promise to exploit the wealth of complex natural forms, to realize novel freeform devices that surpass the art of mainstream biomimetics, and impact critically on crossdisciplinarity nanotechnologies.

## 5. Patents

Methods reported are the subject of patents pending.

## Figures and Tables

**Figure 1 biomimetics-09-00487-f001:**
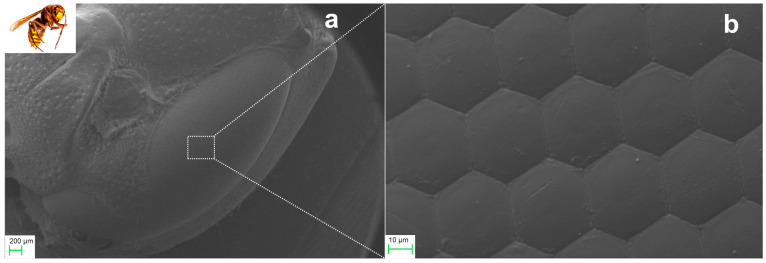
SEM images of natural eyes of Vespa Crabro Flavofasciata: (**a**) Frontal view of the head. (**b**) Facets of compound eye cornea.

**Figure 2 biomimetics-09-00487-f002:**
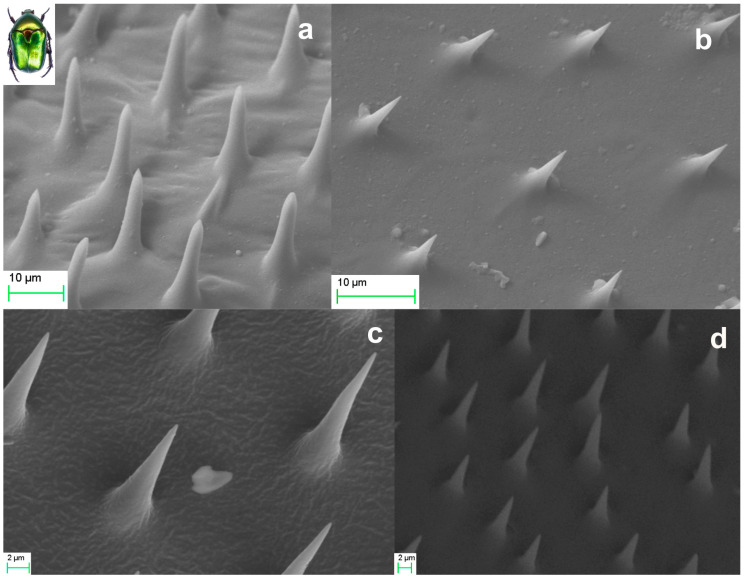
SEM images of natural microtrichia of *Protaetia cuprea phoebe*: Microtrichia on different areas (**a**,**b**) of dorsal side of elytron. Hindwing microtrichia from two distinct areas (**c**,**d**).

**Figure 3 biomimetics-09-00487-f003:**
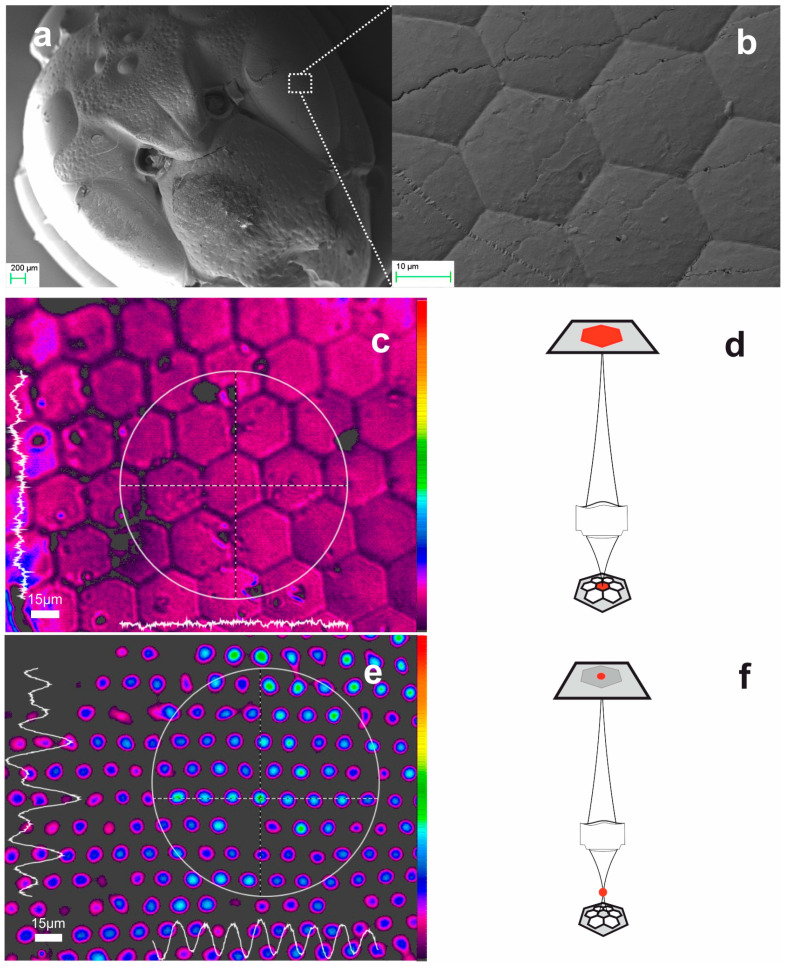
Replication of hornet’s head in aerogel: (**a**) SEM images of aerogel hornet’s head replica (far-view) and (**b**) hornet’s hexagonal microlens array structure. (**c**) microscope image of compound eye surface under the focusing condition of the schematic (**d**). (**e**) Image of focal spots under the focusing condition of the schematic (**f**).

**Figure 4 biomimetics-09-00487-f004:**
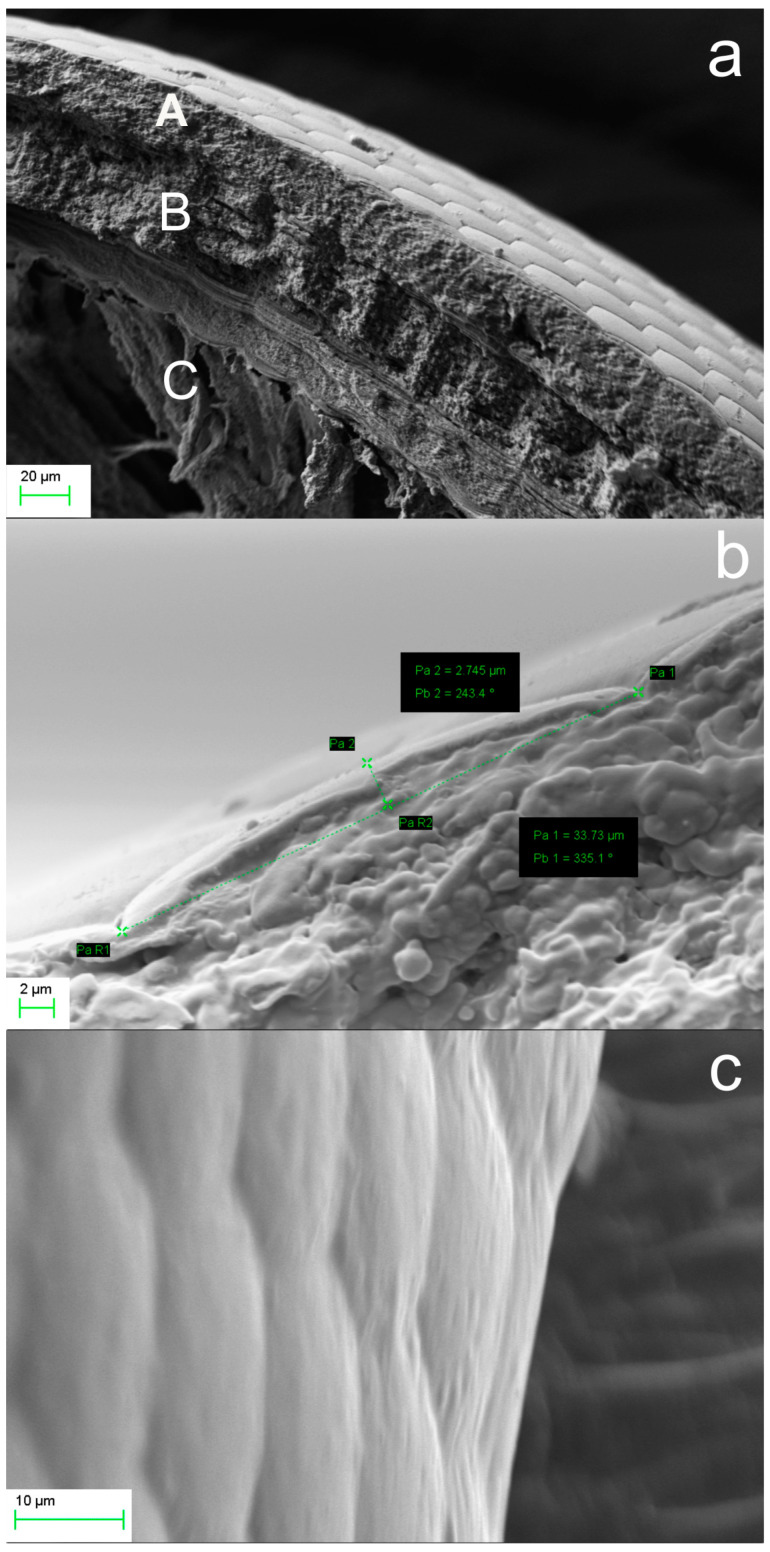
SEM micrographs of: (**a**) cross section of natural compound eye of hornet, where the cornea surface (A), the multilayer structure (B), and the rhabdoms (C); (**b**) close-up view and dimension bars, (**c**) aerogel replica of the compound eye at high inclination.

**Figure 5 biomimetics-09-00487-f005:**
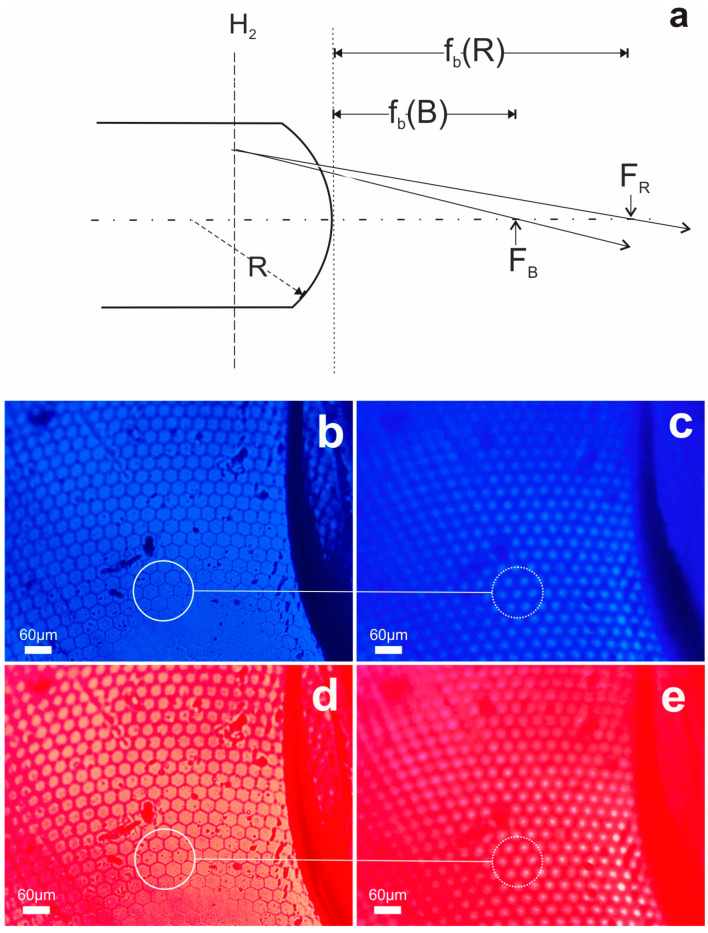
Surface (full circle) and foci images (dotted circle) formed by the thick aerogel compound eye replica using blue (470 nm) and red (630 nm) filtered light. (**a**) Schematic raytracing from the second principal plane H_2_ to the focal points for the blue F_B_ and red F_R_. The back focal lengths f_b_(B) and f_b_(R) are also indicated. Image of the surface (**b**) at blue filtered light and the respective foci (**c**) are shown. Image of surface (**d**) and foci (**e**) using red filtered light. Microlens ensembles and the corresponding foci are indicated encircled.

**Figure 6 biomimetics-09-00487-f006:**
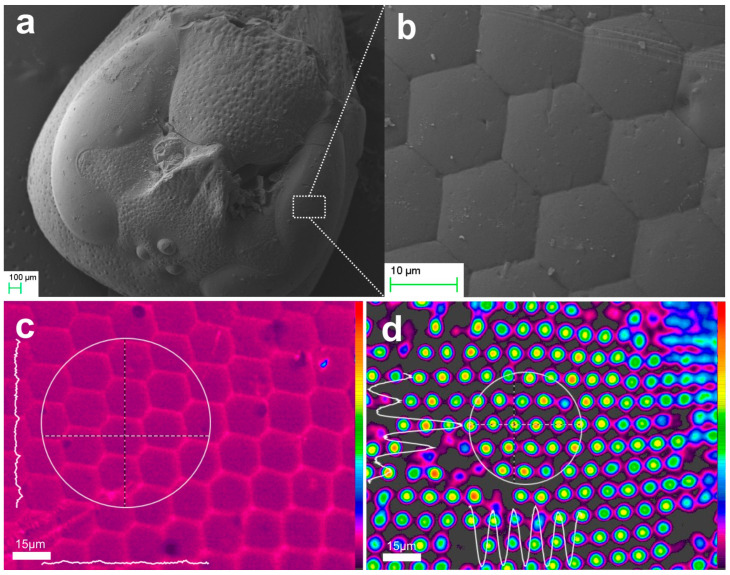
Xerogel replica of hornet’s head: (**a**) SEM images replica (far-view) and (**b**) close-up view of hexagonal microlens array structure. (**c**) microscope image of replica’s surface and (**d**) Image of the focal spots. Microlens ensembles and the corresponding foci in (**c**,**d**) are encircled.

**Figure 7 biomimetics-09-00487-f007:**
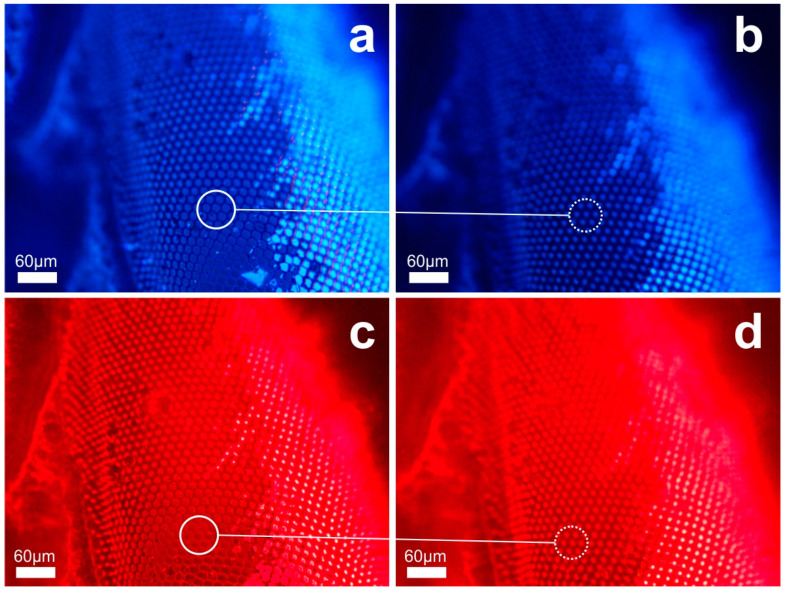
Surface (full circle) and foci images (dotted circle) formed by the thick xerogel microlens element using blue (470 nm) and red (630 nm) filtered light. Surface image (**a**) and foci image (**b**) using blue filtered light. Surface image (**c**) and foci image (**d**) using red filtered light. Microlens ensembles and the corresponding foci are indicated encircled.

**Figure 8 biomimetics-09-00487-f008:**
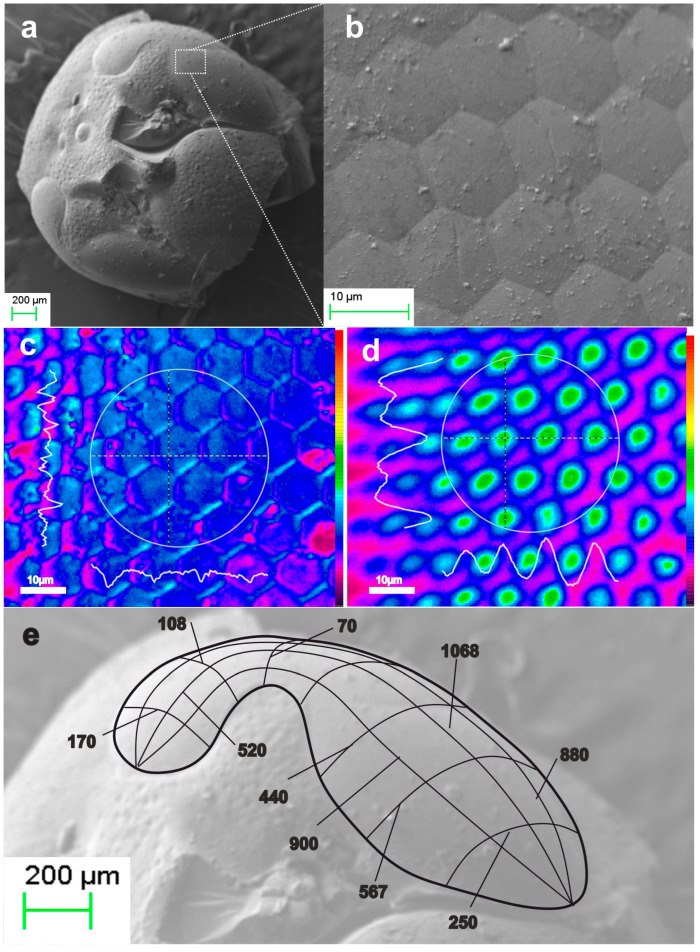
Vitrified hornet’s head replica: (**a**) SEM images replica (far-view) and (**b**) close-up view of hexagonal microlens array structure. (**c**) Microscope image of replica’s surface and (**d**) image of the respective focal spots, by use of the beam profiler. Corresponding areas of microlenses and respective foci in (**c**,**d**) are indicated encircled. (**e**) An anatomical contoured schematic of the compound eye surface superposed on the SEM image (**a**) for illustration. The values of radii of curvature of the representative contours shown are in μm.

**Figure 9 biomimetics-09-00487-f009:**
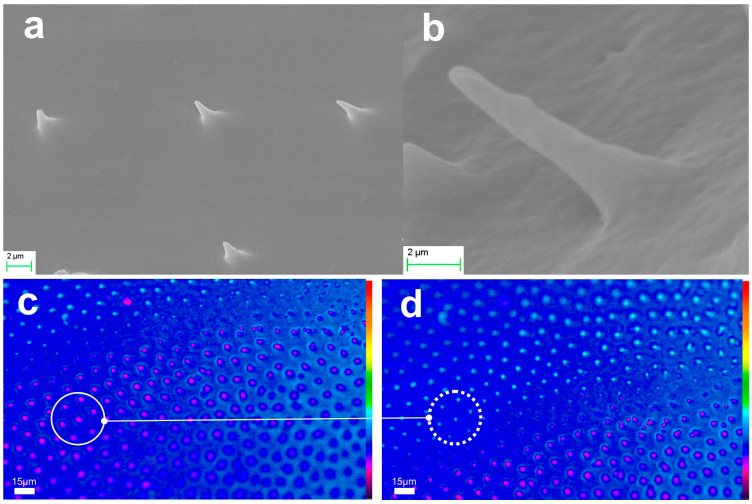
Aerogel replica of elytra microtrichia of the scarab: (**a**) SEM images of aerogel microtrichia array replica (far-view) and (**b**) detail of elytra microneedle structure (close-view). (**c**) microscope image of replica’s microneedles focusing on the cone base (full circle) and (**d**) image focus at the needle apex (dotted circle). Corresponding areas of cone bases in (**c**) and nanotip emission (**d**) are indicated.

**Figure 10 biomimetics-09-00487-f010:**
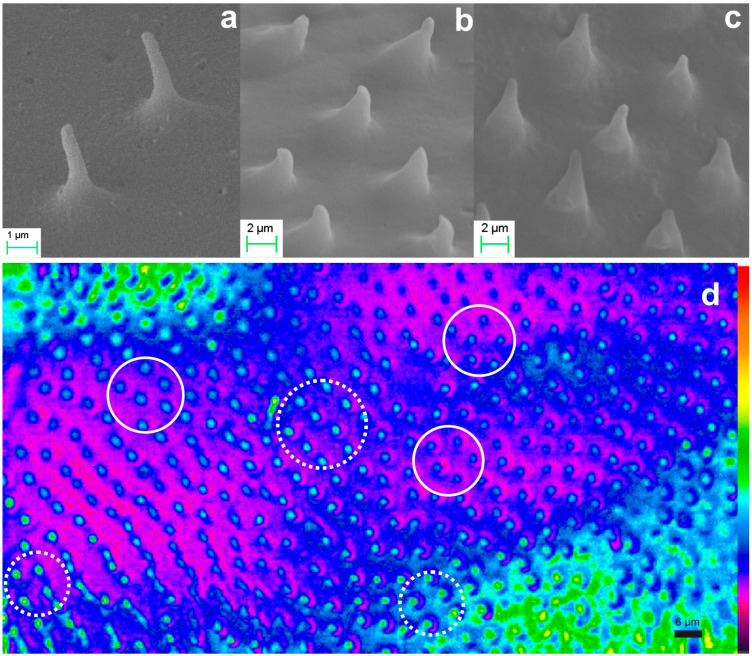
Aerogel replica of hindwing microtrichia of the scarab: (**a**–**c**) SEM images of aerogel microtrichia array replica at three different locations of the wing. (**d**) microscope image of replica’s microneedles focusing on the cone base (full circle) and on the needle apex (dotted circle).

**Figure 11 biomimetics-09-00487-f011:**
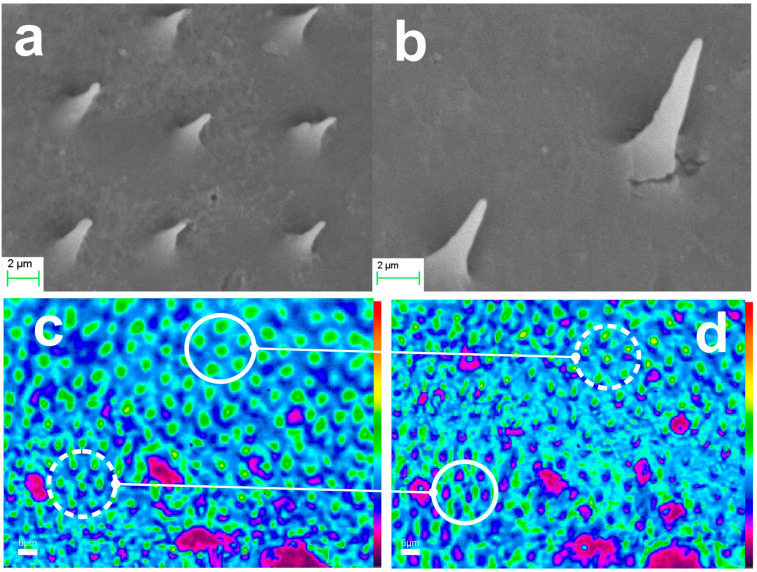
Xerogel replica of elytra microtrichia: (**a**,**b**) SEM images of xerogel microtrichia array replica of two different areas of the elytron. (**c**,**d**) microscope image of replica’s microneedles focusing on the needle cone base (full circle) and at the nanotip apex (dashed circle). Corresponding areas in (**c**,**d**) are indicated.

**Figure 12 biomimetics-09-00487-f012:**
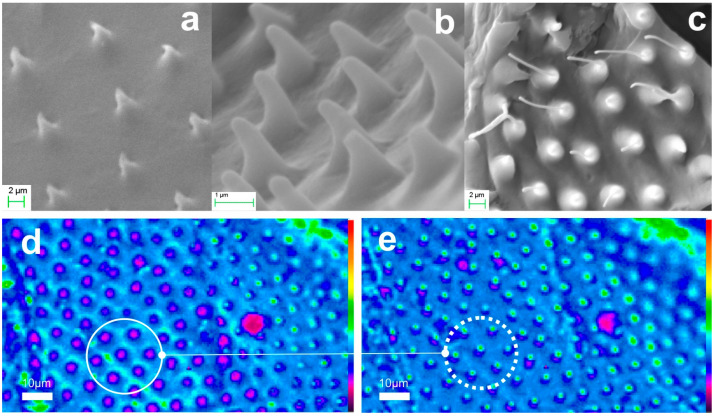
Xerogel replicas of hindwing microtrichia of the scarab: (**a**–**c**) SEM images of xerogel microtrichia array replica at three different areas of the wing. (**d**,**e**) microscope image of replica microneedles focusing at the cone base (full circle) and at the needle apex (dotted circle). Corresponding areas in (**d**,**e**) are indicated.

**Figure 13 biomimetics-09-00487-f013:**
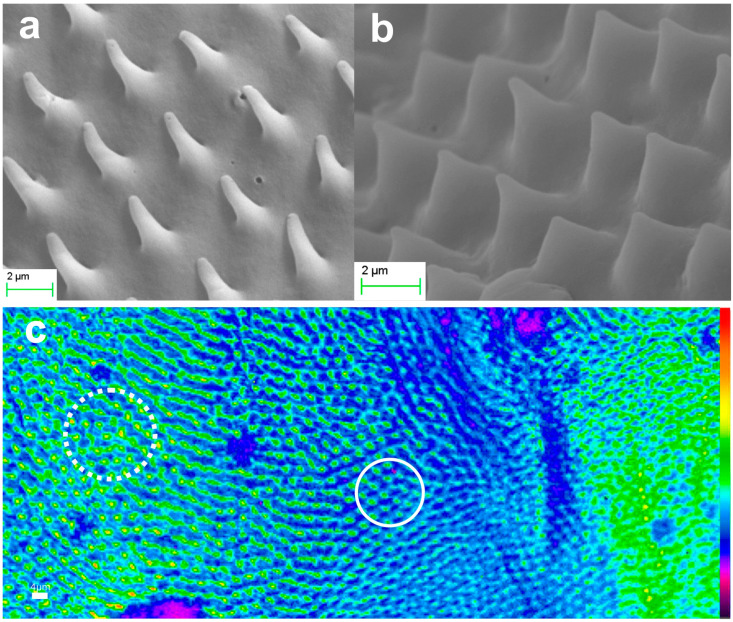
Vitrified xerogel replica of scarab hindwing microtrichia: (**a**,**b**) SEM images of sintered microtrichia array replica at two different locations of the wing. (**c**) microscope image of vitreous nanoneedles focusing on the cone base (full circle) and at the nanotip apex (dotted circle). We note that the image focus setting in these two cases is ~3 μm apart, a distance smaller than the waviness of the surface, and thus only parts of the surface can be set at sharp image focus.

## Data Availability

Relevant data is available upon request.

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
