# Peer review of "Bioarchitectonic Nanophotonics by Replication and Systolic Miniaturization of Natural Forms"

_biomimetics, 2024, doi:10.3390/biomimetics9080487_

Round 1
Reviewer 1 Report
Comments and Suggestions for Authors
In this experimental work the authors present their approach to the fabrication of 3D replicas of samples of insect tissue, namely the arrays of ommatidia (compound eye microlens structures) of European hornet Vespa Crabro Flavofasciata and microtrichia (microneedle array structures) of the golden beetle scarab Protaetia Cuprea Phoebe. These replicas are made in aerogels or xerogels and then downsized 1.5-5 times using the approach the authors previously presented in two other journal articles (they call their shrinkage method systolic miniaturization). For some reason, the authors denoted this whole process as bioarhitectonics, claiming that it represents “progress beyond conventional biomimetics” [sic]. They further characterize the obtained structures, regretfully only by SEM and multispectral optical microscopy, but do not perform measurements that are de facto standard for artificial biomimetic photonic elements and are crucial for their multichannel operation.
The novelty and advantages of the presented approach over the state of the art are unclear and a number of unsubstantiated and sometimes contradicting and incorrect claims are made throughout. English grammar and style are quite good and the text is easy to follow, but its organization would greatly benefit from substantial refinement.
In the further text some scientific comments on the presented work are given in point-by-point manner.
REVIEWER’S POINT-BY-POINT COMMENTS
1. After starting Introduction by writing about “nanotechnology in the sub-10nm regime” and “diffraction of light and particle waves” and thus leading the reader to expect that the contents are connected with these topics, the authors continue to write about their experimental structures which are non-diffractive and dedicated to Abbe-limited classical optics (microlensing and array waveguiding).
2. The state of the art only relates for a small part with the further presented research. In it, the authors write “Current technology [1] employs two-dimensional (2D) planar processing methods to build 3D devices on planar substrates.” Whole fields of nanofabrication are completely skipped. When (needlessly) mentioning top-down and bottom-up methods, the self-assembly approach, which represents the core of bottom-up nanofabrication is not even mentioned. Further, a major part of the state of the art should have been dedicated to its subject, namely the fabrication of artificial replicas of biological parts and the current problems related to it, then following by their proposal how to overcome these and its major advantages. It is the opinion of the reviewer that the state of the art must be rewritten by taking all approaches relevant for nanofabrication of photonic elements into account (no cherry-picking !)
3. The novelty and advantages of the paper against the state of the art must have been better explained. To this purpose, the authors should have included into their consideration both various publications on replica-based and fully artificial biomimetic structures by other teams, but also their own previous papers dedicated to systolic miniaturization of aerogels and xerogels. Especially important for readers is to consider the novelty against the authors’ own ref. [7], but also against [6], [19] and [20], and attempt to prove that their contributions are not incremental.
4. The spectacular but unsubstantiated announcements like “ultimate device miniaturization” and many more such claims are unnecessary. It is unclear why the authors claim “Differentiating from the reported biomimetic technologies, we present, for the first time to our knowledge, a distinctly different approach focusing on optics and photonics”, further writing that their “artificial photonic elements are made by exact replication of biological forms”. The latter is a very standard and often used approach in biomimetics. For instance, different methods of exact replications of ommatidia have been reported for about last two decades, including different journal articles, but also even a 2010 master of science thesis. Biomimetic fabrication of microneedle arrays is much less often met in literature, and the reason is that vastly more accurately dimensioned arrays are designed from zero and are routinely produced by standard MEMS technologies, so that their biomimetic replication and further miniaturization represent needless additional mid-steps. The only noticeable difference between the present approach and those conventionally used is the choice of material (aerogels and xerogels), but this is incremental approach only; the authors themselves published articles on the use of these materials for this purpose as journal articles.
5. The authors denote their approach as “bio-architectonics”; however, this term is already utilized in different fields to denote a number of different methodologies, basically none of which coincide with that of the present article. The most often used field related with that term nowadays is that of molecular biology and metabolism engineering, closely followed by morphological description of different living tissues, while in the commercial world the term is used to describe buildings in conventional architecture with structural or functional details mimicking living tissues. The term is even sometimes used in philosophy. As far as this reviewer knows, the term “bio-architectonics” has been used at least since early 1980-ties, denoting the name of an institution. It is unclear why the term is applied here to denote a procedure for biomimetic photonics, since as far as the reviewer is aware the authors are the only persons to use the already existing (albeit relatively rare) term in that new sense. The reviewer is unaware that the authors even offered their own direct definition of the term in this manuscript.
6. The authors are insistent that their approach extends beyond biomimetics, without proving such an extraordinary claim – while actually it fully belongs to the standard biomimetic photonics.
7. Aerogels are known for their low values of refractive index. It is unclear why the authors are sure that these materials are convenient for microlensing and array waveguiding, because optical materials with higher refractive index would be much more convenient for downsizing since the focal lengths and basically the whole systems would be smaller if such materials were used.
8. Accurate optical inter-element coupling is of prime importance for arrayed photonic structures, yet the paper does not even include this crucial segment into its experimental considerations, in spite of its role in multi-focal beam delivery. The same is valid for microimaging and near-field sensing. Actually, none of these important topics are even considered theoretically.
9. The relation between the claim “In this work, we used the dimensional tools provided by the SEM and inferred an experimental error σSEM=±2nm” (lines 124-125) and the claim “The nanotip apex radii are estimated in the sub-50nm range, being clearly unresolved by our SEM instrument” (lines 471-472) is unclear and appears inconsistent. Especially problematic is the second claim since it contradicts well-known standard SEM performance.
10. The only text about the key subject of the article, namely the replica fabrication methodology is compacted in slightly more than one line of text (“The fabrication of 3D nano-sculptured aerogel replicas required the development of alternative soft-lithographic methods”, line 174), while the synthesis of aerogel/xerogel material itself is described in detail in at least 33 lines. Regardless if this omission is connected with the authors’ patent pending, the article is missing its key part and thus it is incomplete and its scientific accuracy cannot be properly assessed.
11. Some topics totally unrelated with the central subject of biomimetic replication of living tissues are quoted in the text (e.g. for aerogel fabrication “it has been successfully used in visual arts” [sic]).
12. The author claim that “Ongoing optimization efforts are underway aiming to maximize the systolic factor and replication fidelity using optical quality materials, with prime focus on nanophotonic devices” (246-248). However, the performance of the presented structures is far from nanophotonics and is actually even above Abbe limit.
13. The claims about the advantages of making replicas over making fully artificial structures starting from scratch the authors call it “exact replication”) are extremely unclear in the light of the difference between Fig.1b (original) and Fig. 3b (replica), since it appears that the quality of the replica is vastly worse than the original, with numerous cracks and damages not existing in the original that far surpass the claimed nanometer precision needed for nanophotonics. Yet, in the manuscript such results are described as “remarkable structural uniformity and reproduction fidelity.”
14. There are non-elucidated contradictory claims within the text between e.g. “preserved the original 3D stereometric form with extremely high fidelity, enabling their exact replication in aerogels and xerogels.” (lines 176-178) and “The natural variance and the plurality of microlenses do not allow exact correspondence and direct comparisons among the elements under study”. (lines 275-77).
15. The following claim about the visible light microscopy is unclear: “The apparent focal spot observed by the microscope is below ~2μm, clearly near the resolution limits of the instrument” (lines 444-445).
16. The methods presented in the paper are denoted as “solutions towards the ultimate device miniaturization” (line 33), while in reality they improve downsizing at maximum only between 1.5 and 5 times, which retains the most part of the structures in the micrometer range, which is very far from modern results in the photonics field (for “ultimate miniaturization” see e.g. literature on metasurfaces and their diffractive metalenses with sub-nm resolutions, including articles about “picophotonics”, but also most recent publications on biomimetic photonics). One should remember that the price to pay for the very modest results presented in the current paper (which are extremely far from “ultimate miniaturization”) are very low refractive index and high fragility and generally poor mechanical and structural properties.
17. Regarding manuscript organization, a fundamentally systematic approach would be essential, especially regarding the consideration of desirable optical properties (some of which are completely skipped). Deletion of certain SEM micrographs with similar parameters would help make the material more concise. Generally, a fundamental reconsideration of the presented results would be beneficial. The advantages of the fabricated structures over the state of the art must be clearly presented (not in principle, but empirically and quantitatively given; this is especially of importance for dimensions, wavelengths, resolutions, alignment data and other parameters of relevance for photonics applications). Finally, the authors should be aware that this review has presented only illustrative examples of the problems with the manuscript.
Author Response
RESPONSE TO REVIEWER #1
We are grateful to the reviewer for the critical but constructive comments. Deeply respecting his/her expert opinion, we appreciate the opportunity given to clarify many points and improve our work.
We would like to address all points raised, aiming to lift all concerns and reservations of the Reviewer.
OUR RESPONSE TO REVIEWER’S GENERAL COMMENTS
Comment
In this experimental work the authors present their approach to the fabrication of 3D replicas of samples of insect tissue, namely the arrays of ommatidia (compound eye microlens structures) of European hornet Vespa Crabro Flavofasciata and microtrichia (microneedle array structures) of the golden beetle scarab Protaetia Cuprea Phoebe. These replicas are made in aerogels or xerogels and then downsized 1.5-5 times using the approach the authors previously presented in two other journal articles (they call their shrinkage method systolic miniaturization).
Reply
We appreciate the reviewer’s understanding since we have been developing this complex methodology for some time. The term “systolic” is etymologically correct and is set to stress the conformality of the contractive action (systole) and differentiate from the common – and usually arbitrary - “shrinkage of things”. The term has not been challenged before either in the press or in front of live audiences.
Comment
For some reason, the authors denoted this whole process as bioarhitectonics, claiming that it represents “progress beyond conventional biomimetics” [sic].
Reply
We deeply respect the opinion and critical position of the Reviewer, but we are obliged to explicate the terminology in literal terms by referring to contents of prestigious dictionaries.
“Bioinspiration” and “biomimetics” is an antique conception and an art found in architecture such as, for example, the columns of Ionic order (shellfish spirals) and Korinthian order (acanthus leaves), and the modernism of Gaudi’s tree-trunks and human skeletons and beyond to present days.
“Architectural” is literally the “one’” of “having or conceived of as having an overall unified design, form, or structure”. We consider the “ones” produced via biological development are the said “Bioarchitectural”.
The imitation of Bio-architectures or biological processes in engineering, and in technical invention in general, is “Biomimicry”.
In our work we DO NOT “imitate” Bioarchitectures or biological processes, but we use their forms or shapes and transform them to functional devices that produce independent and different operations. These devices ARE NOT necessarily imitating or reproducing the biological processes of the master (mother) forms or shapes we have used. We included here the classical paradigm of “compound eye” in our work for completeness and comparison. We demonstrate how different versions of the same form, but of different size and refractive index, can be manufactured and how they operate. This is the reason that we measure and analyze their characteristics. Furthermore the natural exactness of element as a whole has not been presented before.
I addition to the above we demonstrate a totally new function of optically transparent microtrichia replicas. This is totally irrelevant to the biological functions of microtrichia. Many more examples exist.
We hope that the reviewer agrees that our standpoint justifies the sentence: “progress beyond conventional biomimetics”.
“Architectonic” is literally an object “having an organized and unified structure that suggests an architectural design”. “Architectonics” is the science and the art associated with it.
The reader can find in open literature terms such as: “Molecular Architectonics”, “DNA Architectonics”, “Bioarchitectonics” and even “Solid-Vapor Architectonics”, etc. in various relevant contexts.
In our work we apply exact replication and conformal transformation methods to construct functional objects “having an overall unified design, form, or structure” of Bioarchitectural nature. We, therefore, strongly believe that these are “Bioarchitectonic” structures and the science and art of making and studying them is “Bioarchitectonics”.
Comment
They further characterize the obtained structures, regretfully only by SEM and multispectral optical microscopy, but do not perform measurements that are de facto standard for artificial biomimetic photonic elements and are crucial for their multichannel operation.
We regret to disagree with the Reviewers’ opinion. Our devices are very complex 3-dimensional miniature structures, and, to our knowledge, SEM imaging is the best means to investigate their form. The reviewer may have misunderstood the optical experiments we present in the paper. They are NOT plain optical microscopy observations, but they represent experimental functionality characterizations of the fabricated devices. We use these results to deduce performance values and properties based on these measurements.
The Reviewer’s statement on “de facto” standard methods and “multichannel operation” is unclear, and a relevant clarification would have been highly appreciated.
Comment
The novelty and advantages of the presented approach over the state of the art are unclear and a number of unsubstantiated and sometimes contradicting and incorrect claims are made throughout.
We regret to fully disagree because the reviewer’s comments on “unsubstantiated and contradicting claims” are not justified.
Our work demonstrates a novel class of artificial functional miniaturized photonic devices and their experimental operation. They are exact replicas of bioarchitectural forms and distinctly different to conventional biomimetics in terms of (a) materials, (b) methods and (c) operations.
To elaborate and alleviate any confusion we have now added several references and make a longer introductory presentation.
Comment
English grammar and style are quite good and the text is easy to follow, but its organization would greatly benefit from substantial refinement.
We thank the Reviewer for the positive comments and reiterate on changes and refinements we have now made in the revised text and highlighted by coloration.
OUR RESPONSE TO REVIEWER’S #1 POINT-BY-POINT COMMENTS
Comment
- After starting Introduction by writing about “nanotechnology in the sub-10nm regime” and “diffraction of light and particle waves” and thus leading the reader to expect that the contents are connected with these topics, the authors continue to write about their experimental structures which are non-diffractive and dedicated to Abbe-limited classical optics (microlensing and array waveguiding).
Reply
Our reference to “sub-10nm nanotechnology” is a very introductory and rather general widely acceptable statement. The reference to “the diffraction of light and particle waves” limits the lithographic resolution is again a general comment on the well-known physical phenomenon hindering resolution.
In our work we fabricate miniaturized structures that are exact replicas of natural compound-eye forms having very small focal lengths and working distances. They distinctly differ from microlens arrays fabricated by lithography or self-assembled microspheres. Furthermore, the transparent nano-needles delivering light are conformally downsized replicas of natural microtrichia. The paper presents and experimentally characterizes these structures being all very relevant to our introductory discussion.
Comment
- The state of the art only relates for a small part with the further presented research. In it, the authors write “Current technology [1] employs two-dimensional (2D) planar processing methods to build 3D devices on planar substrates.” Whole fields of nanofabrication are completely skipped. When (needlessly) mentioning top-down and bottom-up methods, the self-assembly approach, which represents the core of bottom-up nanofabrication is not even mentioned. Further, a major part of the state of the art should have been dedicated to its subject, namely the fabrication of artificial replicas of biological parts and the current problems related to it, then following by their proposal how to overcome these and its major advantages. It is the opinion of the reviewer that the state of the art must be rewritten by taking all approaches relevant for nanofabrication of photonic elements into account (no cherry-picking !)
We appreciate the Reviewer’s concerns and thankfully accept all suggestions made. We have now enriched the introduction with further references and discussion. In addition, we would like to clarify the following points with a scope to prevent any possible confusion to the reader:
- Our reference [1] is an excellent and recent overview of the available technologies.
- In line 35-36 (original manuscript) the terms “bottom-up and top-down” are used here in a generic context as suited to lithographic processing. Please note that this sentence refers clearly to “lithographic approaches” as stated in text.
- The “self-assembly” methods referenced by the Reviewer are indeed ‘bottom-up methods’ but are NOT the only ones. Self-assembly is used with colloidal systems, organics and rarely nanocrystals to form ordered lattices. Indeed, self-assembled templates have been used to fabricate biomimetic microlens arrays, photonic band gap structures and the like. Such methods, however, are not relevant to the core of our work. Most importantly, the reader may be confused and associate them with nanoparticle assembly in the aerogels. Nevertheless, we would like to cover this point and have now added relevant references on templating technology and extended further our discussion for clarity and completeness.
Comment
- The novelty and advantages of the paper against the state of the art must have been better explained. To this purpose, the authors should have included into their consideration both various publications on replica-based and fully artificial biomimetic structures by other teams, but also their own previous papers dedicated to systolic miniaturization of aerogels and xerogels. Especially important for readers is to consider the novelty against the authors’ own ref. [7], but also against [6], [19] and [20], and attempt to prove that their contributions are not incremental.
We have now extended the discussion on the points raised. In our introduction we have included several references and stressed the difference of our work. On the other hand, we have tried to balance and limit the discussion since the field is very wide and this is not a review paper.
As we have stated before we have been developing the ‘systolic’ concepts and methods for some time. This has been a very tedious and elaborate work which involves chemical synthesis of materials, replication, chemical and high temperature thermal processing to form fully transparent solid volume and its nanostructured details that are conformal to the ‘master’ aerogel object which is an identical replica of the mother natural specimen.
We hope that the Reviewer is an experimentalist, and we trust that he/she realizes the complexity of the operations.
In our Ref. 19, we have demonstrated the potential of tailoring an aerogel bulk by laser beams, densified it conformally to fused silica while preserving the surface and embedded void microstructures. This is our first demonstration proving the potential for the current developments.
In our Ref. 20, we have gone beyond by casting and minifying the whole objects with accuracy and fidelity. We certainly have neither invented the “casting”, nor the “shrinkage of things”, but we do claim this specific ‘systolic’ fabrication methodology using aerogels and xerogels. In that work we demonstrated the potential to achieve “super-resolution fabrication”. In practice a small structural detail can be made smaller. A line of 1.0mm width can be reduced to 250μm width and a line of 1.0μm can be reduced to 250nm, the 100nm to 25nm and so on, and up to the limits set by the nature of materials.
Our Refs. 6 and 7 are both preliminary conference presentations focusing on materials processing methods suitable to realize bioarchitectural solid replicas.
Here we do not focus the fabrication but on the demonstration of the optical and photonic functionalities.
We trust that the Reviewer appreciates that such optics and photonics have not been presented and recognizes the core of this work which is the experimental demonstration and the characterization of miniature photonic elements.
Comment
- The spectacular but unsubstantiated announcements like “ultimate device miniaturization” and many more such claims are unnecessary.
Reply
The Reviewer has misunderstood this rather general introductory sentence in line 34.
This is NOT OUR CLAIM but a plain reference to the indisputable synergy of materials and methods.
We have referred in our response to Comment 3 to the concept of super-resolution fabrication: That is to fabricate beyond the resolution of the original patterning method.
To eliminate the Reviewer’s concerns, we have removed this sentence.
Comment
- It is unclear why the authors claim “Differentiating from the reported biomimetic technologies, we present, for the first time to our knowledge, a distinctly different approach focusing on optics and photonics”, further writing that their “artificial photonic elements are made by exact replication of biological forms”.
Reply
We do not claim “casting” which is an ancient method, nor the “casting of natural elements” making biomimetics.
Our work differs because (a) we replicate bioarchitectures to produce functional objects which DO NOT necessarily mimic the operations of the mother natural elements used for replication (b) we use materials not used before in this approach and apply relevant systolic methods to transform and minimize conformally these bioarchitectural replicas.
The use of compound eye is a very common typical example we have also used here in order to underline the differences.
We would be grateful to the Reviewer citing any works demonstrating this fabrication of aerogel-xerogel-and solid fused silica exact replicas of a whole wasp -or other- insect head, insect wings, microtrichia etc. and the optical and photonic functionalities of such micro and nanoelements.
Comment
- The latter is a very standard and often used approach in biomimetics. For instance, different methods of exact replications of ommatidia have been reported for about last two decades, including different journal articles, but also even a 2010 master of science thesis. Biomimetic fabrication of microneedle arrays is much less often met in literature, and the reason is that vastly more accurately dimensioned arrays are designed from zero and are routinely produced by standard MEMS technologies, so that their biomimetic replication and further miniaturization represent needless additional mid-steps. The only noticeable difference between the present approach and those conventionally used is the choice of material (aerogels and xerogels), but this is incremental approach only; the authors themselves published articles on the use of these materials for this purpose as journal articles.
Reply
We strongly object to the Reviewer’s unjustified opinion on the “needlessness” of our methods. The citation of references of MEMS or any other technology fabricating objects and functions such as, for example, those shown in Fig. 9 or Fig. 13 will be highly appreciated.
We have added several references and extended our discussion in Section 1 and Section 4. Referring to the reply on comment 5 with the hope to have dissolved all concerns.
Comment
- The authors denote their approach as “bio-architectonics”; however, this term is already utilized in different fields to denote a number of different methodologies, basically none of which coincide with that of the present article. The most often used field related with that term nowadays is that of molecular biology and metabolism engineering, closely followed by morphological description of different living tissues, while in the commercial world the term is used to describe buildings in conventional architecture with structural or functional details mimicking living tissues. The term is even sometimes used in philosophy. As far as this reviewer knows, the term “bio-architectonics” has been used at least since early 1980-ties, denoting the name of an institution. It is unclear why the term is applied here to denote a procedure for biomimetic photonics, since as far as the reviewer is aware the authors are the only persons to use the already existing (albeit relatively rare) term in that new sense. The reviewer is unaware that the authors even offered their own direct definition of the term in this manuscript.
Reply
We appreciate the Reviewer’s concerns and refer to our previous reply. Indeed, the term “bioarchitectonic” has been used in relevant contexts as for example in biological chemistry, (Phil. Trans. R. Soc. B 372: 20160387), without, however, deviating from its principal meaning. As the Reviewer spotted it had also been used as the title of a University Unit.
Bearing in mind that the term “architecture” and its derivatives are widely used in many different contexts, technologies and schools of thought, we are convinced that these other notations cannot challenge the use of the term “Bioarchitectonic” in our work.
Comment
- The authors are insistent that their approach extends beyond biomimetics, without proving such an extraordinary claim – while actually it fully belongs to the standard biomimetic photonics.
Reply
We believe that our previous discussion has justified our point and lifted the Reviewer’s concerns. Additions and clarifications are now made in text to support our point.
Publishing this work in a core “Biomimetics” journal corroborates that our work falls within the said broad “Biomimetics” family. However, we have proved our claims and maintain our position that our methods demonstrate a distinctly new category of devices.
Comment
- Aerogels are known for their low values of refractive index. It is unclear why the authors are sure that these materials are convenient for microlensing and array waveguiding, because optical materials with higher refractive index would be much more convenient for downsizing since the focal lengths and basically the whole systems would be smaller if such materials were used.
Reply
Aerogels are ultraporous materials formed of a solid skeleton filled with air (or other gas). Ultrahigh porosity is needed to achieve systolic downsizing. There exist no materials having such a high porosity simultaneously with a high refractive index. In our work we use silica as a standard material to demonstrate the concept, although others would also produce appreciable results. Certainly, focal lengths are long, waveguide confinement is weak, but mass and heat transport are minute. Such benefits are now stressed in the revised text.
Comment
- Accurate optical inter-element coupling is of prime importance for arrayed photonic structures, yet the paper does not even include this crucial segment into its experimental considerations, in spite of its role in multi-focal beam delivery. The same is valid for microimaging and near-field sensing. Actually, none of these important topics are even considered theoretically.
Reply
The suggested point is beyond the scope of this work. Here we fabricate and study novel photonic elements. Interfacing and application will be the subject of further work.
Comment
- The relation between the claim “In this work, we used the dimensional tools provided by the SEM and inferred an experimental error σSEM=±2nm” (lines 124-125) and the claim “The nanotip apex radii are estimated in the sub-50nm range, being clearly unresolved by our SEM instrument” (lines 471-472) is unclear and appears inconsistent. Especially problematic is the second claim since it contradicts well-known standard SEM performance.
Reply
Our SEM Zeiss EVO MA 10 has not a field-emission but a LAB6 source. The experimental error is provided by the SEM tool and refers to planar well-prepared surfaces that have quite uniform conductivity. Here we image a nanotip of unknown curvature and conductivity. It is impossible to get a clear close-up sharply focused image of the tip with 2nm resolution. This uncertainty justifies is now clarified better in the manuscript.
Comment
- The only text about the key subject of the article, namely the replica fabrication methodology is compacted in slightly more than one line of text (“The fabrication of 3D nano-sculptured aerogel replicas required the development of alternative soft-lithographic methods”, line 174), while the synthesis of aerogel/xerogel material itself is described in detail in at least 33 lines. Regardless if this omission is connected with the authors’ patent pending, the article is missing its key part and thus it is incomplete and its scientific accuracy cannot be properly assessed.
Reply
We extended this discussion adding details to follow the suggestion.
Comment
- Some topics totally unrelated with the central subject of biomimetic replication of living tissues are quoted in the text (e.g. for aerogel fabrication “it has been successfully used in visual arts” [sic]).
Reply
This introductory statement is very relevant as it concerns aerogel applications. An interesting sculpturer and visual artist may use our work in his/her art.
Comment
- The author claim that “Ongoing optimization efforts are underway aiming to maximize the systolic factor and replication fidelity using optical quality materials, with prime focus on nanophotonic devices” (246-248). However, the performance of the presented structures is far from nanophotonics and is actually even above Abbe limit.
The Reviewer’s comment “far from nanophotonics and is actually even above Abbe limit” is not relevant.
We reiterate that we do not fabricate perfect optics, but we replicate the natural forms. The fact does not necessarily imply that the fabricated microlens or the nanotip are perfect. The natural form may not be perfect by some human criteria.
Having characterized fully the fabricated devices and deducing the M2-factor – a parameter
absent in the majority of published biomimetics works, we focus our further work on the optimization of materials and methods towards nanophotonics applications.
Comment
- The claims about the advantages of making replicas over making fully artificial structures starting from scratch the authors call it “exact replication”) are extremely unclear in the light of the difference between Fig.1b (original) and Fig. 3b (replica), since it appears that the quality of the replica is vastly worse than the original, with numerous cracks and damages not existing in the original that far surpass the claimed nanometer precision needed for nanophotonics. Yet, in the manuscript such results are described as “remarkable structural uniformity and reproduction fidelity.”
We strongly disagree with the points raised.
The Reviewer does not make any reference to criteria of comparison and also misses the important point: Fig 3 (b) depicts an ultra-porous aerogel object (>95% porosity) produced under high temperature and high pressure supercritical drying conditions.
Indeed, there is a microcrack in the top left side of image Fig. 3(b), which proves that the figure depicts an original aerogel replica and not a polymer copy. Furthermore, one can spot micron and submicron size surface blemishes and scratches because the natural specimen used is not perfect.
The object is indeed a remarkable reproduction as we explain in lines 269-277 of the original version: “Figure 3 presents….”.
Comment
- There are non-elucidated contradictory claims within the text between e.g. “preserved the original 3D stereometric form with extremely high fidelity, enabling their exact replication in aerogels and xerogels.” (lines 176-178) and “The natural variance and the plurality of microlenses do not allow exact correspondence and direct comparisons among the elements under study”. (lines 275-77).
We regret to totally disagree with this opinion because these statements are mutually fully compatible as they refer to totally different matters of:
- The stereometry referenced in lines 176-178 and also discussed in the analysis of Fig. 4.
- The One-to-One exactness
We hope that the Reviewer realizes that it is practically impossible to compare a specific ommatidium or microtrichium with their corresponding replicas.
Comment
- The following claim about the visible light microscopy is unclear: “The apparent focal spot observed by the microscope is below ~2μm, clearly near the resolution limits of the instrument” (lines 444-445).
Reply
The observed is a bright point source in free space. It is formed by light guided through and emitted off the nanoneedle apex. The bright spot image formed is indeed at the resolution limits of the instrument.
Comment
- The methods presented in the paper are denoted as “solutions towards the ultimate device miniaturization” (line 33), while in reality they improve downsizing at maximum only between 1.5 and 5 times, which retains the most part of the structures in the micrometer range, which is very far from modern results in the photonics field (for “ultimate miniaturization” see e.g. literature on metasurfaces and their diffractive metalenses with sub-nm resolutions, including articles about “picophotonics”, but also most recent publications on biomimetic photonics). One should remember that the price to pay for the very modest results presented in the current paper (which are extremely far from “ultimate miniaturization”) are very low refractive index and high fragility and generally poor mechanical and structural properties.
Reply
Current planar technologies can fabricate in the sub 10nm scale, and nanowires grown chemically can be again in the sub-50nm scale.
We are addressing here free-form object fabrication in 3D space. The reviewer can refer to the excellent article in Optics Express 28, 2683 (2020), observe the dimensions achieved by state-of-the-art technology and compare them with our work.
On the second point, the Reviewer confuses our work with the emerging field of “picophotonics”, which is related to the “topologically structured light and studies of light-induced picometer-scale phenomena” i.e. temporal and spatial modulation in the pico-scale.
We are happy, however, to reference the recent OPN article by Zheludev and MacDonald (OPN, Sept 2023 p.36). One can spot in p.39 the futuristic 3D artist’s impression of a 3D chevron nanowire array and compare its form and dimensions with the real object we have fabricated and present in our Fig. 13(b).
Comment
- Regarding manuscript organization, a fundamentally systematic approach would be essential, especially regarding the consideration of desirable optical properties (some of which are completely skipped). Deletion of certain SEM micrographs with similar parameters would help make the material more concise. Generally, a fundamental reconsideration of the presented results would be beneficial. The advantages of the fabricated structures over the state of the art must be clearly presented (not in principle, but empirically and quantitatively given; this is especially of importance for dimensions, wavelengths, resolutions, alignment data and other parameters of relevance for photonics applications). Finally, the authors should be aware that this review has presented only illustrative examples of the problems with the manuscript.
We have made substantial revisions to the manuscript and added statements to clarify the above comments. All changes are light cyan highlighted. Two figures are amended.
We strongly believe that we have responded to all points raised by the Reviewer and remain grateful for his/her agreement to publish our work.

Reviewer 2 Report
Comments and Suggestions for Authors
This paper provides a three-dimensional nanofabrication method using biopolymeric porous materials like xerogels and aerogels to fabricate nano-sculptured replicas of natural specimens such as the cornea of hornet Vespa Crabro Flavofasciata and the microtrichia ensemble of scarab Protaetia Cuprea Phoebe. The manuscript is original, interesting, and useful information for the readers. The following are my comments and recommendations:
Abstract:
- The abstract should be re-written more clearly and with more details. Please focus the abstract on your experiment, results and conclusions.
Introduction Section:
- The introduction provides many statements that are not supported by references. References are very scarce. Better citations will help authors to propose more accurate information.
- Lines 82-87: Please remove this paragraph, since it is not part of the instrumentation used in the experiment.
- Lines 113 and 120: In this reference (Ref [8]) there are no SEM images to analyse. The authors may have cited an incorrect reference.
- Lines 137-142: I would put here a reference.
- Figure 4: data in Fig. 4b are not clearly visible. Moreover, the augmentation of the images should be included in the caption.
- Line 318: I suppose that the paraxial focal length of the airgel microlens is not 1117 μm but is 111.7 μm. Please correct.
- Line 318: Please put the FWHM's units.
- Figure 5: Put the meaning of F(b) and F(R) in the figure caption.
- Line 345: There is an error here, it is not Fig. 4(e) but Fig. 5(e)
- Line 392: Please put the radius of curvature's units.
- Line 421: There is an error here, it is not Fig. 9(d) but Fig. 10(d)
- In this work, the Results and Discussion Section is, in reality, the Results Section. A Discussion section is missing, in which the authors must interpret the significance of their findings regarding what was already known about the research concern being investigated. The discussion must explain the obtained results, and if possible, compare the results with the findings from other studies. I encourage the authors to expand their Discussion of the results, they can also include implications, limitations, and recommendations.
Author Response
RESPONSE TO REVIEWER #2.
Comments and Suggestions for Authors
This paper provides a three-dimensional nanofabrication method using biopolymeric porous materials like xerogels and aerogels to fabricate nano-sculptured replicas of natural specimens such as the cornea of hornet Vespa Crabro Flavofasciata and the microtrichia ensemble of scarab Protaetia Cuprea Phoebe. The manuscript is original, interesting, and useful information for the readers.
Reply
We are grateful to the Reviewer for his/her valuable comments and suggestions. We have made all the recommended revisions and hope to have lifted all concerns.
Comment
The following are my comments and recommendations:
Abstract: - The abstract should be re-written more clearly and with more details. Please focus the abstract on your experiment, results and conclusions.
Introduction Section: - The introduction provides many statements that are not supported by references. References are very scarce. Better citations will help authors to propose more accurate information.
We have now made substantial changes and have included many references. We trust that the new version is now acceptable for publication.
Comment
- Lines 82-87: Please remove this paragraph, since it is not part of the instrumentation used in the experiment.
Reply
We clarify that this is part of the instrumentation and materials were used in this work and we retain the sentence.
Comment
- Lines 113 and 120: In this reference (Ref [8]) there are no SEM images to analyse. The authors may have cited an incorrect reference.
Reply
We greatly appreciate the point and have made the correction.
Comment
- Lines 137-142: I would put here a reference.
Reply
Relevant references are now added.
Comment
- Figure 4: data in Fig. 4b are not clearly visible. Moreover, the augmentation of the images should be included in the caption.
Reply
We have amended and enlarged the image to make visible the data.
Comment
- Line 318: I suppose that the paraxial focal length of the airgel microlens is not 1117 μm but is 111.7 μm. Please correct.
Reply
Aerogel is a very rare material with an extremely low refractive index. The focal length is indeed 1117 μm.
Comments
- Line 318: Please put the FWHM's units.
- Figure 5: Put the meaning of F(b) and F(R) in the figure caption.
- Line 345: There is an error here, it is not Fig. 4(e) but Fig. 5(e)
- Line 392: Please put the radius of curvature's units.
- Line 421: There is an error here, it is not Fig. 9(d) but Fig. 10(d)
Reply
We thank very much the Reviewer and have made all corrections.
Comment
- In this work, the Results and Discussion Section is, in reality, the Results Section. A Discussion section is missing, in which the authors must interpret the significance of their findings regarding what was already known about the research concern being investigated. The discussion must explain the obtained results, and if possible, compare the results with the findings from other studies. I encourage the authors to expand their Discussion of the results, they can also include implications, limitations, and recommendations.
Reply
We deeply appreciate all the points raised and recommendations made and have now extended the discussion as suggested. Please observe the revised and more detailed introduction which we believe enhances the concepts of our work.
We thank very much the Reviewer for his/her constructive comments and hope that these modifications make our work acceptable.
All changes are highlighted in light cyan.
Please note changes in Figures.

Reviewer 3 Report
Comments and Suggestions for Authors
This work uses low refractive index aerogel and xerogel to replica the surface of insects and build micro and nano optical devices, which is an interesting stragety for fabrication of micro and nano optical devices. However, there are several major issues about this work:
1. Indeed, the authors make some optical devices by replica molding of insect surfaces, but they did not investigae the surface roughness of the device obtained which is vital for optical devices. Moreover, the repeatability of this fabrication methods also needs to be inverstigated.
2. And in Figure 3(b), there are some cracks on the device surface. In Figure 6(b) and Figure 8(b), there are some small 'dusts' on the device surface. The authors should provide an detailed explaination about these cracks and dusts. Are these cracks and dusts products of fabrication itself or other sources (such as, there exists cracks and dust on the surface of insects)?
3. In the abstract 'whilst transparent nanotips deliver light in the nanoscale', and in the line 489 'their optical performance has been investigated'. I can not agree with the authors, the investigation of the optical performance of these devices obtained is very superficial and not enough for a research paper.
In addtion, I agree this method can be used to fabricate some nano features for optics, but the insect surface is not so uniform (as can be seen from these SEM pictures). The authors should also provide a dicussion about how to ensure consistent performance of such devices.
In the abstract 'Systolic miniaturization using these master forms generates fused-silica super-resolution micro/nanodevices', 'forms generates' should be 'forms'?
Author Response
RESPONSE TO REVIEWER 3
Comments and Suggestions for Authors
This work uses low refractive index aerogel and xerogel to replica the surface of insects and build micro and nano optical devices, which is an interesting stragety for fabrication of micro and nano optical devices. However, there are several major issues about this work:
Reply
We are grateful to the Reviewer for appreciating our work and submitting very valuable comments which we have fully considered.
We would like to reply to the specific points raised as follows.
Comment
- Indeed, the authors make some optical devices by replica molding of insect surfaces, but they did not investigate the surface roughness of the device obtained which is vital for optical devices. Moreover, the repeatability of this fabrication methods also needs to be investigated.
Reply
Surface roughness is a function of (a) the material used i.e. aerogel is rougher than the xerogel and xerogel is rougher than densified replica (b) the surface quality of the original natural specimen and (c) the process itself which any case is interrelated to the above. We have included roughness values of samples. Certainly, the methods are not optimized as one can observe, for example, in the submicron blemishes of the fully vitrified replica of Fig. 8(b). The high repeatability of fabrication is a matter of process automation on which we are now extending our effort.
Comment
- And in Figure 3(b), there are some cracks on the device surface. In Figure 6(b) and Figure 8(b), there are some small 'dusts' on the device surface. The authors should provide an detailed explaination about these cracks and dusts. Are these cracks and dusts products of fabrication itself or other sources (such as, there exists cracks and dust on the surface of insects)?
Reply
The observed microcracks are due to the aerogel fabrication process and it is indeed the proof that this object is a highly porous aerogel and not a polymer copy. Please note that is the product of high-pressure high-temperature supercritical drying of a gel. One can also observe submicron scale blemishes due to the imperfections of the natural specimen itself, used as the master object.
Comment
- In the abstract 'whilst transparent nanotips deliver light in the nanoscale', and in the line 489 'their optical performance has been investigated'. I cannot agree with the authors, the investigation of the optical performance of these devices obtained is very superficial and not enough for a research paper.
Reply
We appreciate the concern of the Reviewer but would like to underline that these objects are prototype alternative devices made of non-standard materials. They are parts of larger 3D replicas of very complex stereometry and not planar structures. To our knowledge, their stereometry, the minute dimensions and their physical nature do not allow much room for further investigations, such as, for example, the measurement of aberrations. However, we have studied experimentally important morphological and the optical characteristics of these devices under white light and monochromatic RGB illumination. Critical parameters have been deduced starting from the back-focal length of compound eye microlenses, the principal plane location, their focal lengths, the FWHM of the beamlet i.e. the infinite conjugate ratio PSF, and the M2 factor (the latter representing a prime parameter of beam-quality which is absent in most biomimetics works), as well as measures of chromatic dispersion for the materials used. In addition, we have demonstrated experimental nanotip performances and have measured relevant parameters where possible.
Comment
In addtion, I agree this method can be used to fabricate some nano features for optics, but the insect surface is not so uniform (as can be seen from these SEM pictures). The authors should also provide a discussion about how to ensure consistent performance of such devices.
Reply
We have extended relevant discussions and added references in the introduction.
Comment
In the abstract 'Systolic miniaturization using these master forms generates fused-silica super-resolution micro/nanodevices', 'forms generates' should be 'forms'?
Reply
At this point we mean “forms” in the context of “shapes” i.e master (mother) object shapes. We have now revised the sentence to alleviate this confusion.
We thank very much the Reviewer for his/her constructive comments and hope that these modifications make our work acceptable.
All changes are highlighted in light cyan. Please note changes in Figures.

Round 2
Reviewer 1 Report
Comments and Suggestions for Authors
The authors worked diligently and made a large number of modifications of their original manuscript. The result is that the novel version is much better than the previous one. However, the main problem with the manuscript novelty still exists, and after reading the clarifications by the authors it remains unclear how and if it can be corrected. This reviewer’s opinion, clarified in the below point-by-point comments, is that the contribution of the research, although performed investing quite a lot of work and effort, is incremental (as observed from the point of view of the already published solutions).
In addition to the above essential problem, a number of problematic claims from the original version still remain in the revised manuscript, although many of others are now corrected. Some new problems appear in the newly introduced modifications. It must be also said that besides the mentioned problematic claims in the revised manuscript, the reply to the reviewer introduces some new ones. As in the previous review, some scientific comments and suggestions of the reviewer are given below in point-by-point manner.
Reviewer’s Point-By-Point Comments
1. The novelty of the revised manuscript still remains unclear, even after large modifications. It is incremental at best, compared to the already existing publications. The adequacy of the approach for micro-optical applications is questionable (see also the next point).
2. According to the authors, probably the most important advantage of aerogel/xerogel-based optical applications is that they represent “ultralightweight refractive optical elements”. In reality their low weight is of rather unclear practical importance for micro-optics, and the used approach is paid by 1) high proneness to mechanical damages and cracking of these porous biomimetic structures and 2) their extremely low effective refractive index. Both of these properties are very inconvenient for optical applications.
3. There are erroneous claims in the authors’ cover letter. The authors write for their structures “They are exact replicas of bioarchitectural forms and distinctly different to conventional biomimetics in terms of (a) materials, (b) methods and (c) operations.” Actually, contemporary biomimetics includes replicas made with a plethora of different materials, they are being done by a number of methods and used for operations unrelated to the natural structure which was mimicked/replicated. Even some of the papers written by this reviewer present experimental biomimetic approaches belonging to all three cases (including super-resolution fabrication). The journal rules strongly discourage quoting the reviewer’s references and this is the reason why I do not quote any of them. Instead, to ensure a general view of works of others, I simply suggest the authors to use Google Scholar and do a search for “biomimetic ommatidia” or “biomimetic microlens arrays” for their first topic and “biomimetic microneedle arrays” for the second topic. Another good source is (obviously) Scopus. This advice is also related with the request related to the topic #6.
4. This reviewer claims that all of the results presented as bioarchitectonic ones belong to contemporary biomimetics and that the used approach represent only a small subset of optical biomimetics. What’s more, the approach described in the manuscript offers performance below the standard biomimetics and is thus unclear why is it presented as a significant novelty. The fields of extreme interest nowadays are nanophotonics/near field optics that vastly exceed the performance of the micro-optics. On the other hand, the results described by the authors offer micro-optical far-field operation whose performance and resolutions are severely limited by the porous nature of the aerogel materials.
5. In their cover letter the authors requested the reviewer to cite any works demonstrating the “fabrication of aerogel-xerogel-and solid fused silica exact replicas of a whole wasp -or other- insect head, insect wings, microtrichia etc. and the optical and photonic functionalities of such micro and nanoelements.” The reviewer is obliged to say that due to the above mentioned shortcomings of the use of aerogel-xerogel for microlenses and microneedles these materials are rightfully almost nonexistent in the contemporary research, since they offer vastly poorer and less controllable optical properties compared to the modern structures like metasurfaces, etc., and hardly can even exceed the performance of conventional micro-optical devices. So no, in the opinion of this reviewer the non-presence of the topic throughout biomimetic optical literature in this case does not represent its novelty or originality, but only illustrates practical inconveniences, lab complexities and low performance compared to other techniques. This is the reason why the reviewer does not feel obliged to respond positively to the posed citation request.
6. The authors make a bold claim that systolic method “enables super-resolution fabrication of functional 3D devices having arbitrary, freeform, stereometry, unavailable by any other means to date.” I tend to disagree with that, since optical near fields are often used to achieve super-resolutions which can be in principle much higher than those described in the article (again: the internal structure of aerogels or xerogels – the pores size – may seriously limit the achievable resolutions).
7. The reviewer disagrees with the authors’ argumentation related with some of the points (e.g. 15, 16, 18 in their cover letter). For instance, regarding point 18 the authors cite a 5 years old article from Optics Express related with PBG where one encounters pores with radius of about 160 nm (pores in PBG are related solely with the operating wavelength and the chosen material) as an illustration of “dimensions achieved by state-of-the-art technology,” which is an incorrect claim, since the similar PBG dimensions have been obtainable literally decades ago. In the same point (18) the authors also describe an image from a Sep 2023 OPN article by Zheludev & MacDonald and compare their structures to it. Actually the OPN illustration presents an ultraprecise chevron nanowire-based metamaterial while the authors’ article analyzes arrays of microneedles integrated with a solid surface and sticking from it so that the two figures are almost completely unrelated both structurally and from the points of view of the utilized material and technology. The authors also write “the Reviewer confuses our work with the emerging field of ‘picophotonics’.” Actually nothing is confused there and picophotonic arrays are mentioned solely to illustrate the extreme accuracy of contemporary high-precision nanophotonic structures where some of them are able to reach functionality at subnanometer level, contrasted to micro-optical structures not even enabling one to surpass the Abbe limit.
8. In the newly added text (lines 665, 666) the authors claim “bioarchitectonic concepts and methods promise to exploit the wealth of complex natural forms and enable novel free-form 3D devices and functionalities beyond the art of mainstream biomimetics” although the modern biomimetics includes what they call bioarchitectonics as one of its subsets. In line 635 a nanoneedle is mentioned.
9. Some small errors are made in lines 529, 530 where the signs [] appear in a few places – probably the authors intended to enter ref numbers and accidentally missed it.
Comments on the Quality of English Language
English is exceptionally good and there are only some minor corrections to be made, mostly contained within the newly added revisions in version 2 of the manuscript.
Author Response
Dear Reviewer 1,
The authors thank Reviewer 1 for their valuable comments and suggestions, all of which have been considered. The authors have requested the Editor to terminate the review process.
Yours sincerely,
Konstantina Papachristopoulou

Reviewer 2 Report
Comments and Suggestions for Authors
The authors have adequately addressed the reviewer's comments thoroughly and satisfactorily. However, I have a couple of comments to make:
- Line 117: this is the first time that the abbreviation (PDMS) is mentioned in the text, please put its meaning ''polydimethylsiloxane (PDMS)''.
- Line 146: please remove ''(3)''.
- Lines 529, 530, and 531: What do the brackets ''[ ]'' mean here? Maybe the authors forgot to put some references?
- Line 541: There is an error here, it is not Fig. 7(d) but Fig. 8(d).
Comments on the Quality of English Language
Some minor grammatical mistakes should be revised.
Author Response
Reviewer 2
We are grateful to the Reviewer for the invaluable recommendations, which we have taken into account below.
Comments and Suggestions for Authors
The authors have adequately addressed the reviewer's comments thoroughly and satisfactorily. However, I have a couple of comments to make:
Reply
We are pleased to have adequately addressed the previous comments and suggestions from the Reviewer. We have now made all the recommended revisions.
Comment
Line 117: this is the first time that the abbreviation (PDMS) is mentioned in the text, please put its meaning ''polydimethylsiloxane (PDMS)''.
Reply
We have added the full name of the PDMS.
………………………………………………………………………………………………………………………………………………………………….
Comment
Line 146: please remove ''(3)''.
Reply
We have removed the element ''(3)''.
………………………………………………………………………………………………………………………………………………………………….
Comment
Lines 529, 530, and 531: What do the brackets ''[ ]'' mean here? Maybe the authors forgot to put some references?
Reply
Indeed, there are missing references in the brackets. Relevant references have been added.
………………………………………………………………………………………………………………………………………………………………….
Comment
Line 541: There is an error here, it is not Fig. 7(d) but Fig. 8(d).
Reply
We have made the suggested amendment.
………………………………………………………………………………………………………………………………………………………………….
Comments on the Quality of English Language
Some minor grammatical mistakes should be revised.
Reply
Grammatical mistakes have now been revised.
………………………………………………………………………………………………………………………………………………………………….
We thank very much the Reviewer for his/her constructive comments and hope that these modifications make our work acceptable.

Reviewer 3 Report
Comments and Suggestions for Authors
1. For the reply of my first comment, the authors say 'The high repeatability of fabrication is a matter of process automation on which we are now extending our effort.', I suggest the authors to resubmit the manuscript after they get more data about the fabrication repeatability.
2. For the reply of my second comment, does the microcracks caused in the aerogel fabrication process have an effect on the optical properties of the microdevice? The authors are suggested to clarify it.
Author Response
Reviewer 3
We are grateful to the Reviewer for the invaluable recommendations, which we have taken into account below.
Comment
For the reply of my first comment, the authors say 'The high repeatability of fabrication is a matter of process automation on which we are now extending our effort.', I suggest the authors to resubmit the manuscript after they get more data about the fabrication repeatability.
Reply
The fabrication method we employed is a technique developed through research and is continuously evolving and improving. Indeed, our experiments show good repeatability at a research laboratory level. We are not an industry and thus cannot report the highest repeatability that can be seen in industrial processes nor do we have the resources to run long-term repeatability testing.
…………………………………………………………………………………………………………………………………………………………………
Comment
For the reply of my second comment, does the microcracks caused in the aerogel fabrication process have an effect on the optical properties of the microdevice? The authors are suggested to clarify it.
Reply
The defects are at the submicron level, contributing only to some additional light scattering effects. As demonstrated in Fig. 3(e) and Fig. 5(b-e), these microcracks do not affect critically the optical performance of the microoptical element.
…………………………………………………………………………………………………………………………………………………………………

Round 3
Reviewer 1 Report
Comments and Suggestions for Authors
The last version of the submitted manuscript does nothing to improve its rather incremental and even weak novelty and does not reply to any of the crucial questions posed, starting from the incompatibility of its notation with the standard and conventional one used in modern biomimetics, extending over the incompatibilities among the text contents, etc. It is the opinion of this reviewer the text does nor satisfy the strict and highly demanding requirements of the Biomimetics journal. Obviously, the final decision has to be made by the Editors.
Author Response
Response to Reviewer 1
We are grateful to Reviewer 1 for his/her constructive comments and recommendations which we have taken into account.
We really appreciate the criticism which made us have a deeper view in our work and make revisions that we believe will be useful to the scientific audience.
Nevertheless, we feel deeply disappointed because Reviewer 1 did not recognize the novelty of the concepts and the efforts this work has taken.
Regarding the negative opinions we have fully documented the rightness of our statements, methods and results in our latest response to the Academic Editor in Round 2. It is rather pointless to reiterate.
Yours sincerely,
Konstantina Papachristopoulou
Reviewer 2 Report
Comments and Suggestions for Authors
The authors have satisfactorily addressed my comments. Therefore, I have one additional comment for the authors to consider:
- Line 305: Ref. [75] has been repeated twice [7575], please correct.
Author Response
Response to Reviewer 2
We are grateful to Reviewer 2 for his/her constructive comments and recommendations. We have made all suggested revisions and we greatly appreciate the final recommendation of Revier 2 for publishing our work. Please note that the error in Line 305: Ref. [75] has been corrected.
Yours sincerely,
Konstantina Papachristopoulou
Reviewer 3 Report
Comments and Suggestions for Authors
This version is good for publication now.
Author Response
Response to Reviewer 3
We are grateful to Reviewer 3 for his/her constructive comments and recommendations. We have made all suggested revisions and we greatly appreciate the final recommendation of Revier 3 for publishing our work.
Author Response
Response to Academic Editor
Dear Academic Editor,We resubmit the manuscript with minor missing references and corrections as suggested. We have replied to Reviewer 2 and Reviewer 3 and have made the suggested changes. In view of the above we will not reply to Reviewer 1. However, for deontological reasons we attach in Annex below our response to Reviewer 1.
Reviewer 1 Comments in italic font
Author Response in plain font
The authors worked diligently and made a large number of modifications of their original manuscript. The result is that the novel version is much better than the previous one. However, the main problem with the manuscript novelty still exists, and after reading the clarifications by the authors it remains unclear how and if it can be corrected. This reviewer’s opinion, clarified in the below point-by-point comments, is that the contribution of the research, although performed investing quite a lot of work and effort, is incremental (as observed from the point of view of the already published solutions).
In addition to the above essential problem, a number of problematic claims from the original version still remain in the revised manuscript, although many of others are now corrected. Some new problems appear in the newly introduced modifications. It must be also said that besides the mentioned problematic claims in the revised manuscript, the reply to the reviewer introduces some new ones. As in the previous review, some scientific comments and suggestions of the reviewer are given below in point-by-point manner.
Authors Response
Even though Reviewer 1 recognizes the effort invested in this work, he/she insists on disregarding the contents of our paper.
Vaguely the Reviewer 1 states again that our work is already published elsewhere, without however providing any evidence to support his/her opinion. Probably the Reviewer considers that our prior work and our prior knowledge on this subject understates the importance of this paper. We have acknowledged in our text that these results are the products of long-term research and accumulated know-how.
If Reviewer 1 had found our results published elsewhere then he/she must have cited the relevant reference and must have firmly rejected our paper.
Reviewer 1 repeats his/her vague quotations on “problematic claims” we are making without providing any justification.
For the above reasons it is very unclear.
Regarding the specific Reviewer’s Point-By-Point Comments :
- The novelty of the revised manuscript still remains unclear, even after large modifications. It is incremental at best, compared to the already existing publications. The adequacy of the approach for micro-optical applications is questionable (see also the next point).
Authors Response
Reviewer 1 questions the novelty and the adequacy of the approach without providing any clear evidence or justification to support his/her position.
We consider that the quality of this peer review lacks any scientific merit. We do not accept it and we strongly object.
- According to the authors, probably the most important advantage of aerogel/xerogel-based optical applications is that they represent “ultralightweight refractive optical elements”. In reality their low weight is of rather unclear practical importance for micro-optics, and the used approach is paid by 1) high proneness to mechanical damages and cracking of these porous biomimetic structures and 2) their extremely low effective refractive index. Both of these properties are very inconvenient for optical applications.
Reviewer 1 misjudges the development of “ultralightweight refractive optical elements” on rather premature grounds by stating “rather unclear practical importance of micro-optics”.
Authors Response
Our experimental results prove that such structurally rare ultraporous materials can indeed have refractive optical functionality. This is an experimental research result and not the validation of a commercial product. Furthermore, the reviewer overlooks the fact that-as we present in the paper- we can achieve stepwise densification and consequent strengthening of the material. Furthermore, we can have other types of aerogels made of high strength oxides and carbides, plus we can include additives and reinforce the skeleton at minimal expense of functionality.
Is the Reviewer able to prove that today’s research results will not have any practical importance in the future?
- There are erroneous claims in the authors’ cover letter. The authors write for their structures “They are exact replicas of bioarchitectural forms and distinctly different to conventional biomimetics in terms of (a) materials, (b) methods and (c) operations.” Actually, contemporary biomimetics includes replicas made with a plethora of different materials, they are being done by a number of methods and used for operations unrelated to the natural structure which was mimicked/replicated. Even some of the papers written by this reviewer present experimental biomimetic approaches belonging to all three cases (including super-resolution fabrication). The journal rules strongly discourage quoting the reviewer’s references and this is the reason why I do not quote any of them. Instead, to ensure a general view of works of others, I simply suggest the authors to use Google Scholar and do a search for “biomimetic ommatidia” or “biomimetic microlens arrays” for their first topic and “biomimetic microneedle arrays” for the second topic. Another good source is (obviously) Scopus.
Authors Response
This advice is also related with the request related to the topic #6.
Once again, Reviewer 1 quotes on our “erroneous claims”! without any plausible justification.
Reviewer 1 is suggesting to us to do a library search!
We have done a very detailed search and have cited all relevant works we have found in all scientific databases including Scopus. Thus, we have cited eighty (80) works in the revised version which are in the biomimetics, nanophotonics and nanofabrication fields, many of them are not necessarily closely relevant to our work, but they are just included in text as a background, for illustrating the subjects and comparing the technologies.
Definitely, the claim of the Reviewer that somebody else has previously published OUR work is untrue. The Reviewer’s position is clearly unjustified and lacks scientific merit.
Furthermore, hiding behind the words of “journal rules strongly discourage quoting the reviewers references” is rather opaque. The Journal does not enforce an obligation, but only makes a recommendation. Expressing such strong opinions without evidence is totally unacceptable and against all academic ethics.
- This reviewer claims that all of the results presented as bioarchitectonic ones belong to contemporary biomimetics and that the used approach represent only a small subset of optical biomimetics. What’s more, the approach described in the manuscript offers performance below the standard biomimetics and is thus unclear why is it presented as a significant novelty. The fields of extreme interest nowadays are nanophotonics/near field optics that vastly exceed the performance of the micro-optics. On the other hand, the results described by the authors offer micro-optical far-field operation whose performance and resolutions are severely limited by the porous nature of the aerogel materials.
Authors Response
The Reviewer position is totally undocumented.
The Reviewer either does not understand what is the paper about or intentionally presents vague arguments in order to understate our work : What is “performance below the standard biomimetics”…!!! Biomimetics can be macro-, and micro- and nano -optics…!
Finally, in the last two sentences of Comment 4 the Reviewer totally overlooks (1) that the nanotips deliver light in the near-field and (2) that the performances are not limited by the porous nature – this is what the paper is about- and not only this, the paper densifies the porous materials and forms optics made of pure fused silica !
- In their cover letter the authors requested the reviewer to cite any works demonstrating the “fabrication of aerogel-xerogel-and solid fused silica exact replicas of a whole wasp -or other- insect head, insect wings, microtrichia etc. and the optical and photonic functionalities of such micro and nanoelements.” The reviewer is obliged to say that due to the above mentioned shortcomings of the use of aerogel-xerogel for microlenses and microneedles these materials are rightfully almost nonexistent in the contemporary research, since they offer vastly poorer and less controllable optical properties compared to the modern structures like metasurfaces, etc., and hardly can even exceed the performance of conventional micro-optical devices. So no, in the opinion of this reviewer the non-presence of the topic throughout biomimetic optical literature in this case does not represent its novelty or originality, but only illustrates practical inconveniences, lab complexities and low performance compared to other techniques. This is the reason why the reviewer does not feel obliged to respond positively to the posed citation request.
Authors Response
Fortunately, we are the first to develop such methodologies and we believe it will be noteworthy publishing them in Biomimetics.
The unjustified claims of the Reviewer lack any scientific merit, since he/she cannot realize - or intentionally pretends not to realize - that what we present here is an alternative technology. We do not repeat the work of others.
- The authors make a bold claim that systolic method “enables super-resolution fabrication of functional 3D devices having arbitrary, freeform, stereometry, unavailable by any other means to date.” I tend to disagree with that, since optical near fields are often used to achieve super-resolutions which can be in principle much higher than those described in the article (again: the internal structure of aerogels or xerogels – the pores size – may seriously limit the achievable resolutions).
Authors Response
The Reviewer cannot realize - or intentionally pretends not to realize- our concept and the methodology of super-resolution we present here.
The Reviewer believes that super-resolution is ONLY achieved by near field optics! Certainly he/she bares in mind that super-resolution is ‘the subwavelength’ and ONLY this.
We have repeated and explained in several points in text that: the Systolic approach takes any structure fabricated by near-field optics, electron beams or ion beams and makes it smaller. This is the super-resolution nanofabrication we mean and its limits are only the natural limits of the material we use.
- The reviewer disagrees with the authors’ argumentation related with some of the points (e.g. 15, 16, 18 in their cover letter). For instance, regarding point 18 the authors cite a 5 years old article from Optics Express related with PBG where one encounters pores with radius of about 160 nm (pores in PBG are related solely with the operating wavelength and the chosen material) as an illustration of “dimensions achieved by state-of-the-art technology,” which is an incorrect claim, since the similar PBG dimensions have been obtainable literally decades ago. In the same point (18) the authors also describe an image from a Sep 2023 OPN article by Zheludev & MacDonald and compare their structures to it. Actually the OPN illustration presents an ultraprecise chevron nanowire-based metamaterial while the authors’ article analyzes arrays of microneedles integrated with a solid surface and sticking from it so that the two figures are almost completely unrelated both structurally and from the points of view of the utilized material and technology. The authors also write “the Reviewer confuses our work with the emerging field of ‘picophotonics’.” Actually nothing is confused there and picophotonic arrays are mentioned solely to illustrate the extreme accuracy of contemporary high-precision nanophotonic structures where some of them are able to reach functionality at subnanometer level, contrasted to micro-optical structures not even enabling one to surpass the Abbe limit.
Authors Response
First, the Reviewer does not realize -or pretends not to realize - that the Optics Express article cited is a work concerning 3-dimensional PBG in the short wavelength region. If the Reviewer had spotted another 3D PBG with smaller features, we would be more than happy to cite it.
Second, the Reviewer does not realize or pretends not to realize, that “picophotonics” relates to structured light fields. In the article Sep. 2023 OPN article by Zheludev & MacDonald the reader finds:
- “The convergence of pioneering work on topologically structured light and studies of light-induced picometer scale phenomena have laid the foundations of a new and dynamic discipline” (page 37 of OPN Sep. 2023)
- “The picometer scale precision achieved in these recent optical measurements reaches far beyond the spatial resolution of conventional optical microscopes” (page 38 of OPN Sep. 2023)
- “Artists view of a chevron nanowire metamaterials “ in inset legend of figure on page 39 and the ~1μm scale-bar measuring 2mm in the printed picture, depicts the Artist’s View of chevron wires having about 150 nm diameter ordered at a period of 450nm. In our previous reply we had kindly asked the Reviewer to have a look and compare with the structures of our Fig. 13(b) in terms of form and dimensions
Obviously, the Reviewer either does not comprehend the science behind this, or pretends not to comprehend it.
In any case, Comment 7 made by the Reviewer is groundless.
- In the newly added text (lines 665, 666) the authors claim “bioarchitectonic concepts and methods promise to exploit the wealth of complex natural forms and enable novel free-form 3D devices and functionalities beyond the art of mainstream biomimetics” although the modern biomimetics includes what they call bioarchitectonics as one of its subsets. In line 635 a nanoneedle is mentioned.
Authors Response
In our previous response to Reviewer 1 we have explained in detail the concepts and the terminology by referring to prestigious dictionaries of Oxford, Cambridge and Merriam-Webster.
We have added such details in section 3.1, from line 338 onwards, of the revised manuscript.
- Some small errors are made in lines 529, 530 where the signs [] appear in a few places – probably the authors intended to enter ref numbers and accidentally missed it. 7
Authors Response
We thank all Reviewers spotting these missing references, which are now added in the revised manuscript uploaded.
Comments on the Quality of English Language
English is exceptionally good and there are only some minor corrections to be made, mostly contained within the newly added revisions in version 2 of the manuscript.
Authors Response
We thank the Reviewer and have now made corrections.
Yours sincerely,
Konstantina Papachristopoulou